# A DIMENSIONAL R2 REGRESSION METRIC

## ABSTRACT

Evaluation metrics are the primary guide in modeling. For regression tasks, the R2 score is the gold standard, offering a magnitude-agnostic measure of accuracy that captures variance. However, R2 has three key limitations: it is limited to at most two dimensional inputs, it reduces the score to a single scalar that hides rich patterns of prediction accuracy, and it is sensitive to low-variance noise channels which can yield large, uninterpretable negative values. We introduce the Dimensional R2 score (Dim-R2), a simple extension of R2 that accepts data of arbitrary dimensionality, provides a multidimensional view of accuracy, and reduces sensitivity to noise. We demonstrate its advantages on both synthetic sinusoidal data and three multidimensional regression data. Dim-R2 offers an interpretable and flexible metric that illuminates patterns in regression accuracy, guiding regression modeling.

## 1 INTRODUCTION

Evaluation metrics are the lighthouse of modeling. They quantify how well the model's predictions match the target data and guides decisions such as model tuning and data cleaning (Jordan & Mitchell, 2015). For regression tasks, the R2 score is considered the gold standard compared to metrics like mean absolute error (MAE) or mean squared error (MSE) (Chicco et al., 2021). MAE shows a simple magnitude based error but cannot differentiate between biased predictions and lazy mean predictions. MSE penalizes variance and resolves the lazy mean prediction issue, but its value ranges per data domain making it hard to interpret. In contrast, R2 is a normalized, data domain-independent metric that reflects variance explained by the model normalized by the variance of the data (Eq. 1). It ranges from 1 (perfect prediction) to $-\infty$, with 0 indicating performance equal to the lazy mean prediction. Due to its ability to capture variance and its normalized score, R2 is widely used in regression evaluation (Chicco et al., 2021; Sykes, 1993; Ash & Shwartz, 1999; Sauerbrei et al., 2020; Pedregosa et al., 2011).

However, R2 has three key limitations. First, R2 is defined for 1D data which is averaged across channels for 2D data, and it cannot be directly applied to higher-dimensional regression data (Heckel et al., 2024; Ahmed et al., 2025; Zhang et al., 2023). Second, the R2 reduces model performance to a single scalar or a 1D score, offering no insight into how accuracy varies across data dimensions (Fig. 10). A multidimensional view of regression accuracy could reveal structure that could help modelers target specific features of their data and model for improvement. Third, R2 is highly sensitive to low-variance noise channels in multi-channel (2D) regression tasks. It can yield large negative values when the true data has little variation, as is the case for noisy channels. When these R2 scores are averaged across channels for multi-channel data, the mean R2 can be largely negative which obscures the presence of high accuracy channels (Fig. 11).

To address these limitations, we introduce the Dimensional R2 score (Dim-R2). Dim-R2 simply flattens selected dimensions into independent observations and computes the standard R2, while retaining the shape of the remaining dimensions. It accepts regression data of arbitrary dimensionality, overcoming the 2D limitation of conventional R2 by flexibly flattening any dimensions. As Dim-R2 can flatten and keep any dimensions, it also enables a multidimensional view of prediction accuracy (Fig. 10). For example, in data shaped (Trials, Time, Channels) (Perich et al., 2020; Yoo et al., 2022), Dim-R2 can reveal data regions that are predictable in certain trials, specific time periods, or certain channels, highlighting noisy trials, temporally localized features, or task-relevant channels. This multidimensional score helps modelers identify patterns in both their data and models. Finally, by flattening selected dimensions into independent observations, high-variance (informative) chan-

Table 1: Comparison of common regression metrics. Abbreviations: R2 = R2 score; MSE = mean squared error; MAE = mean absolute error; D2-AE = D2 absolute error; EV = explained variance; Corr = Pearson correlation.

| Metric | Captures Variance | Captures Bias | Normalized | Reference Baselines |
|---|---|---|---|---|
| R2 | O | O | O | O |
| MSE | O | O | X | X |
| MAE | X | O | X | X |
| D2 AE | X | O | O | O |
| EV | O | X | O | O |
| Corr ($\rho$) | O | X | O | O |

nels outweigh low-variance (noisy) ones, yielding a more robust score than the mean of per-channel R2 scores. This better highlights the presence of high accuracy channels (Fig. 11).

The paper is organized as follows. We introduce Dim-R2 as a simple extension of the conventional R2 metric, designed to support three key improvements. We then present the dimensional view of regression accuracy using Dim-R2, while emphasizing the effects of Dim-R2 arguments on the result. Next, we evaluate the resilience of Dim-R2 by comparing it to conventional Mean-R2. The input data dimensionality is not compared between Dim-R2 and R2 as it is the metric definition rather than a property of the resulting scores. The dimensional-view is demonstrated using a toy sinusoidal data and several multidimensional regression task where the data contains spatial features. The resilience feature is demonstrated with another toy sinusoidal data and a hyperparameter sweep on data-constrained recurrent neural network (DC-RNN) modeling (Perich et al., 2020; Perich & Rajan, 2020) trained to simulate mouse neural activity during a reach-to-grab task. Throughout this paper, the terms dimensions and axes are used interchangeably.

## 2 RELATED WORKS

Regression tasks (Sykes, 1993; Krzywinski & Altman, 2015; Altman & Krzywinski, 2015) aim to estimate a continuous target variable $y$ from input data $x$, with predictions given by $\hat{y} = f(x)$ where $f(x)$ is any model. A principled regression metric in machine learning should satisfy four criteria: (1) capture the variance structure of the target, (2) penalize bias between $y$ and $\hat{y}$,(3) be normalized to allow comparison across data scales, and (4) provide interpretable reference points (e.g., perfect prediction, mean prediction).

The R2 score satisfies all four criteria and remains the standard regression metric in regression tasks. (Choi et al., 2022; Liu et al., 2024; Kogan et al., 2020). It is a negative squared error between $y, \hat{y} \in \mathbb{R}^N$, the ground-truth target and predicted values, normalized by the variance of $y$. It is defined for a single channel data (1D) with $N$ observations (Eq. 1). The value of R2 ranges from $(-\infty, 1]$, where 1 indicates perfect prediction, 0 corresponds to predicting $\bar{y}$, the mean of $y$, and negative values indicate worse performance than predicting $\bar{y}$.

$$R2 = 1 - \frac{RSS}{TSS} = 1 - \frac{\sum_i (y_i - \hat{y}_i)^2}{\sum_i (y_i - \bar{y})^2},$$

(1)

where $i = 1, ..., N$, RSS, TSS refers to observation index, residual sum of squares, and total sum of squares, respectively. In contrast, many commonly used metrics (Hastie et al., 2015) only satisfy a subset of the criteria necessary for regression metric (Table. 1). The equations of Table 1 are described in the Appendix A.1.

## 3 MATERIALS AND METHODS

### 3.1 REGRESSION METRICS: CONVENTIONAL AND DIMENSIONAL R2

In the multi-channel $y, \hat{y} \in \mathbb{R}^{N \times C}$ (2D), R2 score is measured per channel to yield 1D score or averaged to yield the mean R2 (Eq. 2). As a result, the conventional R2 score supports 1D input, or at most 2D input when computing a mean R2 for a single score.

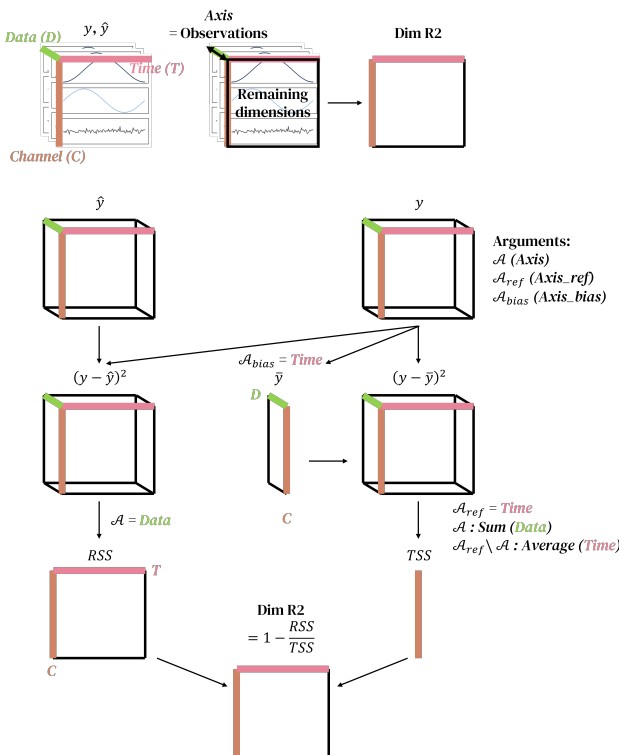

Figure 1: Schematic of Dim-R2 calculation on 3D data. There is one argument $\mathcal{A}$ (Axis) and two optional arguments $\mathcal{A}_{ref}$ (Axis_ref), $\mathcal{A}_{bias}$ (Axis_bias).

$$Mean\ R2 = \frac{1}{C}\sum_{c}^{C}[1 - \frac{\sum_i (y_{i,c} - \hat{y}_{i,c})^2}{\sum_i (y_{i,c} - \overline{y_c})^2}] \tag{2}$$

where $C$, $c$ refers to the number of channels and channel index, respectively.

Dim-R2 follows the concept of Eq. 1 but computes RSS and TSS in multidimensional form (Fig. 1, Eq. 3-6). I takes one argument (Axis) and two optional arguments (Axis_bias, Axis_ref), noted as $\mathcal{A}, \mathcal{A}_{bias}, \mathcal{A}_{ref}$, respectively. Let the full set of axes be $\mathcal{D}$.

$$RSS = \sum_{k\in\mathcal{A}}(y_k - \hat{y}_k)^2, \qquad RSS \in \mathbb{R}^{\mathcal{D}\setminus\mathcal{A}} \tag{3}$$

$$\bar{y} = \frac{1}{|\mathcal{A}_{bias}|}\sum_{k\in\mathcal{A}_{bias}} y_k, \qquad \bar{y} \in \mathbb{R}^{\mathcal{D}\setminus\mathcal{A}_{bias}} \tag{4}$$

$$TSS = \frac{1}{|\mathcal{A}_{ref}\setminus\mathcal{A}|}\sum_{j\in\mathcal{A}_{ref}\setminus\mathcal{A}}\sum_{k\in\mathcal{A}}(y_{k,j} - \bar{y})^2, \qquad TSS \in \mathbb{R}^{\mathcal{D}\setminus(\mathcal{A}_{ref}\cup\mathcal{A})} \tag{5}$$

$$Dim\text{-}R2 = 1 - \frac{RSS}{TSS}, \qquad Dim\text{-}R2 \in \mathbb{R}^{\mathcal{D}\setminus\mathcal{A}} \tag{6}$$

where $k$ refers to indices referring to the data in the specified dimensions. The $|\mathcal{A}_{bias}|$, and $|\mathcal{A}_{ref}\setminus\mathcal{A}|$ indicate the number of observations along the specified axes, used to compute the mean.

$\mathcal{A}$ specifies the axis to collapse, in which the remaining dimensions define the resulting shape of Dim-R2 (Fig. 1). $\mathcal{A}$ is used to sum the error (RSS) and the $y$ variance for normalizing (TSS). $\mathcal{A}_{bias}$ specifies the axis along which $\bar{y}$ is computed, defining the reference level for measuring deviation. It is recommended to keep $\mathcal{A}_{bias}$ small, since a smaller $\mathcal{A}_{bias}$ yields more localized reference levels

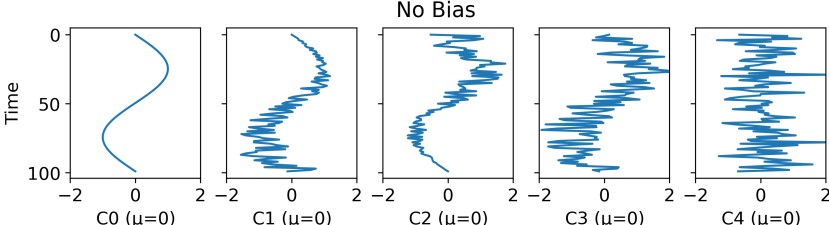

Figure 2: Example sinusoidal waveforms with time-varying noise, used for both $y$ and $\hat{y}$ in showing dimensional accuracy. Different biases per neuron and trial were added to other data – Consistent Neural Bias and Varying Neural Bias.

across the remaining dimensions and therefore provides a more detailed Dim-R2. $\mathcal{A}_{ref}$ specifies the axis along which the reference variance (TSS) is additionally aggregated beyond $\mathcal{A}$. It must include $\mathcal{A}_{bias}$, since the reference variance is computed as deviation from the mean defined along that axis ($\mathcal{A}_{bias} \subset \mathcal{A}_{ref}$). When measuring TSS, averaging is applied to the axes in the set difference $\mathcal{A}_{ref} \setminus \mathcal{A}$ to keep the magnitude of TSS consistent. It is also recommended to keep $\mathcal{A}_{ref}$ small, as a smaller $\mathcal{A}_{ref}$ yields more localized TSS and a more detailed Dim-R2. When $\mathcal{A}_{ref}$ is larger, the normalizing term TSS is shared across $\mathcal{A}_{ref}$ resulting in Dim-R2 with a uniform normalization across $\mathcal{A}_{ref}$. If not specified, $\mathcal{A}_{bias}$ defaults to $\mathcal{A}_{ref}$, and $\mathcal{A}_{ref}$ defaults to $\mathcal{A}$, measuring variance across $\mathcal{A}$.

When computing RSS and TSS, the specified dimensions ($\mathcal{A}$, $\mathcal{A}_{ref}$) are treated as independent observations. Then Dim-R2 is computed by broadcasting the shapes of RSS and TSS. This allows Dim-R2 to accept data of arbitrary dimensionality. Dim-R2 is a generalization of the conventional R2 and reduces to the variance weighted mean R2 under specific axis selections (Appendix A.2). The Dim-R2 is implemented in Python using NumPy (Harris et al., 2020) and follows the scikit-learn syntax (Pedregosa et al., 2011). The implementation is provided in Appendix A.7.3.

### 3.2 DATASETS

#### 3.2.1 SYNTHETIC SINUSOIDAL DATASET

To illustrate the dimensional view of regression accuracy (Section 4.1), we generated a waveform of shape (Data, Time, Channels)=(1000, 100, 5) (Fig. 2). Channel C0 to C3 share the same sine wave while C4 is pure standard Gaussian noise ($\mu = 0, \sigma = 1$) adjusted to have the same variance with clean signal C0. From C0 to C3, the uniform random noise in the range $[-1, 1]$ were added with different temporal patterns: no noise (C0), linearly increasing noise scale from 0 to 1 over time (C1), linearly decreasing noise scale from 1 to 0 over time (C2), and constant noise over time (C3). The $y$ and $\hat{y}$ share the same underlying waveform but different independent random noise, to show the gradual change in accuracy. To illustrate how Dim-R2 arguments affect the dimensional view, two data conditions were used: one with no added bias where time-averaged values are 0 (No Bias), and one with a time bias varying from 0 to 4 across channels C0 to C4, randomly assigned across trials without replacement (Varying Channel Bias) (Fig. 2, 12). The No Bias condition contains no trial variability and minimal cross-channel variability, while the Varying Channel Bias condition contains both trial and cross-channel variability.

To demonstrate how Dim-R2 better reflects the presence of high-accuracy channels compared to mean R2 (Section 4.2), multichannel sinusoidal data of shape (100 timesteps, 100 channels) was generated for $y$ and $\hat{y}$ across 100 random repetitions. A fixed ratio of channels was replaced with Gaussian noise of set variance (Fig. 7). The noise channel variances for $y$ and $\hat{y}$ were set independently as they affect the R2 calculation.

#### 3.2.2 APPLICATION DATASETS

To illustrate the dimensional view of regression accuracy beyond controlled synthetic data, we applied Dim-R2 on several multi-dimensional regression tasks where $y, \hat{y} \in \mathbb{R}^n$ with $n \geq 3$. These include neural activity prediction from DC-RNNs trained on mouse motor cortex Neuropixels recordings, as well as image reconstruction using Variational Autoencoder (VAE) (Pu et al., 2016; Doersch, 2016; Pinheiro Cinelli et al., 2021) on the MNIST (Deng, 2012) and CelebA (Liu et al., 2015)

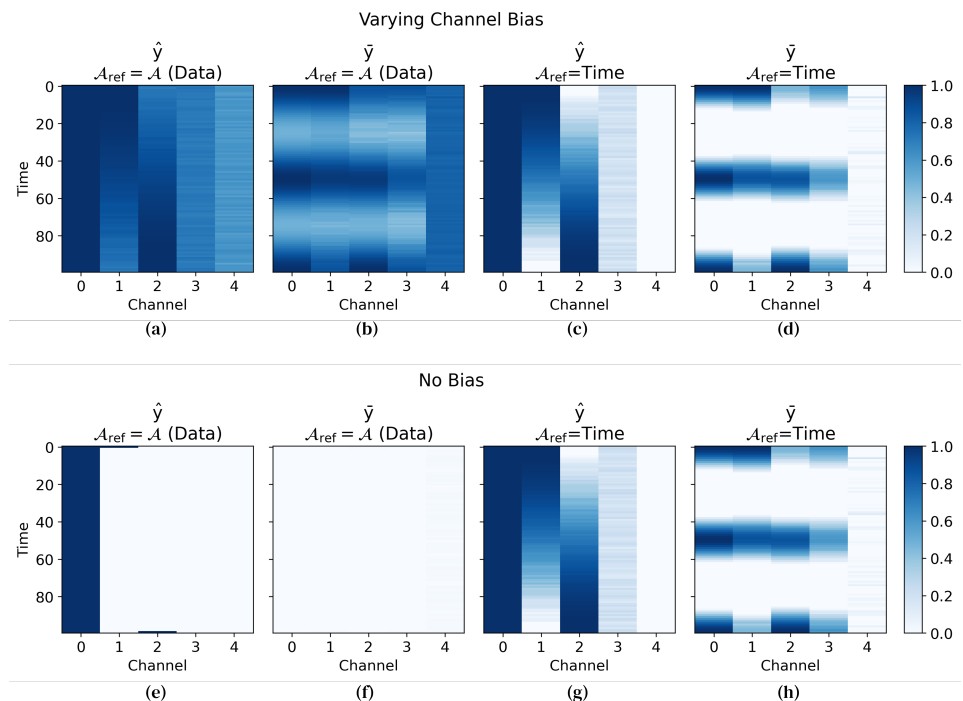

Figure 3: Dim-R2 shows dimensional view of regression accuracy for synthetic sinusoidal dataset. Each heatmap shows Dim-R2 computed between $y$ and the specified prediction, measured across Data dimension to reveal R2 scores across the Time and Channel dimensions. (a) & (e) Dim-R2 on $\hat{y}$ with $\mathcal{A}_{ref} = \mathcal{A}$ (Data). (b) & (f) Dim-R2 on $\bar{y}$ with $\mathcal{A}_{ref} = \mathcal{A}$ (Data). (c) & (g) Dim-R2 on $\hat{y}$ with $\mathcal{A}_{ref}$=Time. (d) & (h) Dim-R2 on $\bar{y}$ with $\mathcal{A}_{ref}$=Time. (a)-(d) Data with varying time bias across data and channels. (e)-(h) Data with no time bias.

datasets. Full details are described in the Appendix A.4 and A.5. These applications contain spatiotemporal features which allow Dim-R2 to reveal how accurately models capture variability along dimensions such as trials, time, channels, and pixels.

To demonstrate the practical value of Dim-R2's noise resilience, we conducted a hyperparameter sweep on the DC-RNN dataset (Appendix A.4), showing how Dim-R2 can guide modelers in selecting hyperparameters despite working with noisy datasets.

## 4 RESULTS

### 4.1 DIM-R2 PROVIDES A DIMENSIONAL VIEW OF REGRESSION ACCURACY

To demonstrate how Dim-R2 provides a dimensional view of regression accuracy, sinusoidal waveforms $y$ and $\hat{y}$ of shape (Data, Time, Channels)=(1000,100,5) with time varying noise were used (Fig. 2). No bias data, and varying time bias across data and channels, were added to show different characteristics of Dim-R2 (Fig. 12). Mean R2 was not measured as it is not defined for data over two dimensions.

Dim-R2 captures different types of variability depending on $\mathcal{A}_{bias}, \mathcal{A}_{ref}$ (Fig. 1). If not specified, $\mathcal{A}_{ref}$ defaults to $\mathcal{A}$, measuring reference variance along the collapsed dimension. For example, setting $\mathcal{A}_{bias} = \mathcal{A}_{ref}$ = Data causes Dim-R2 to set $y$ data variability as its reference (Fig. 3a, c, e, g). When data variability exists in both $y$ and $\hat{y}$, capturing it is a meaningful regression goal. Thus, Dim-R2 yields high scores across all channels including the noise channel 4 (Fig. 3a). While lazy mean prediction $\bar{y}$ across the $\mathcal{A}$ should yield an R2 of zero by definition, time-averaged $\bar{y}$ can still capture data variability since $\bar{y}$ is time-averaged time while $\mathcal{A}_{bias} = \mathcal{A}_{ref}$ is set across data. Thus, Dim-R2 with $\bar{y}$ yields a high score when data variability exists (Fig. 3b), and a near-zero score when it does not (Fig. 3f). When $y$ has very small variance (TSS), R2 amplifies small prediction errors

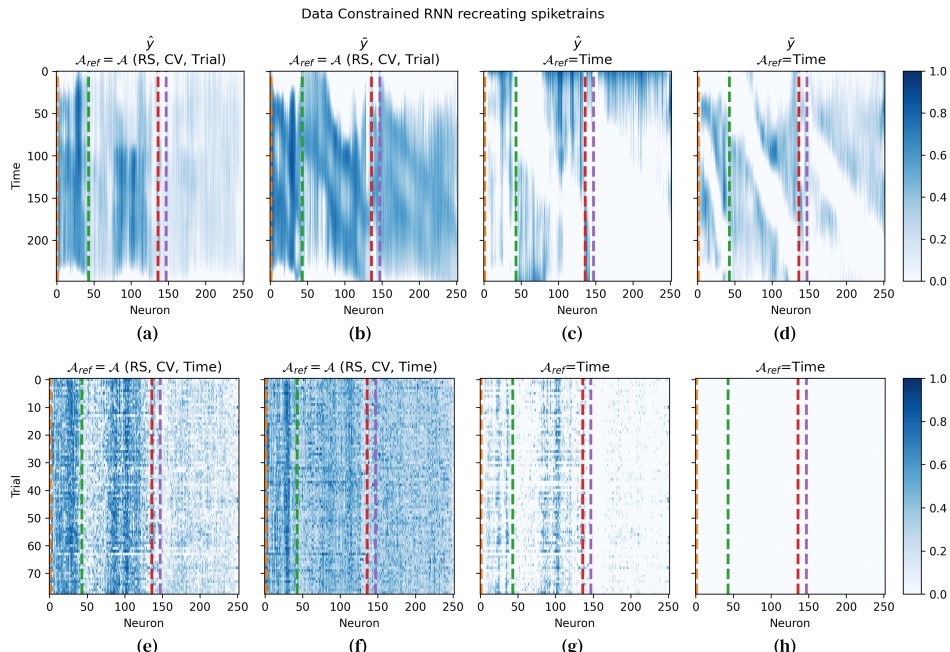

Figure 4: Dimensional view of Dim-R2 for neural data-constrained recurrent neural networks. Each heatmap shows Dim-R2 computed between $y$ and the specified prediction, measured across different dimensions to reveal R2 scores across other remaining dimensions. Dashed lines separate neurons by brain region (from left): DCN (orange), M1 (green), Striatum (red), Thalamus (purple). (a) & (e) Dim-R2 on $\hat{y}$ with $\mathcal{A}_{ref}=\mathcal{A}$. (b) & (f) Dim-R2 on $\bar{y}$ with $\mathcal{A}_{ref}=\mathcal{A}$. (c) & (g) Dim-R2 on $\hat{y}$ with $\mathcal{A}_{ref}$=Time. (d) & (h) Dim-R2 on $\bar{y}$ with $\mathcal{A}_{ref}$=Time. (a)-(d) Dim-R2 with $\mathcal{A}$=(Random seed, Cross validation folds, Trials). (e)-(h) Dim-R2 with $\mathcal{A}$=(Random seed, Cross validation folds, Time).

(RSS), producing large negative scores. The same occurs in Dim-R2 when data variability in $y$ is negligible, where setting $\mathcal{A}_{bias} = \mathcal{A}_{ref}$=Data causes Dim-R2 to yield largely negative values (Fig. 3e).

While capturing data-variability is important, it is also meaningful to assess how well each channel is predicted across time, regardless of data variability. By setting $\mathcal{A}_{bias} = \mathcal{A}_{ref}$=Time and $\mathcal{A}$=Data, the time-varying accuracy of each channel is revealed (Fig. 3c, g). This shifts the reference variance to the time dimension not data, resulting in identical Dim-R2 values for datasets with and without time biases–reflecting only the time-varying noise in $y$ and $\hat{y}$ (Fig. 2).

Note that when $\mathcal{A}_{bias}$ (Time) for measuring $\bar{y}$ is different from $\mathcal{A}$ (Data), Dim-R2 on $\bar{y}$ does not yield 0. This is because R2 equals 0 when $\hat{y} = \bar{y}$ but $\bar{y}$ is measured across $\mathcal{A}_{bias}$ not $\mathcal{A}$. The Dim-R2 for $\bar{y}$ with $\mathcal{A}_{bias} = \mathcal{A}_{ref}$=Time reflects the time-varying error patterns of the sine wave, showing increased scores at the start, middle, and end for Neurons 0 to 3 (Fig. 3d, h). Therefore, interpreting Dim-R2 correctly under different $\mathcal{A}$ and $\mathcal{A}_{ref}$ settings, along with selecting a proper domain-specific negative control such as $\bar{y}$ for reference, is critical for meaningful analysis with Dim-R2. All combinations of $\mathcal{A}, \mathcal{A}_{bias}, \mathcal{A}_{ref}$ for the sinusoidal waveforms are included in the Supplemental File (A.7.1).

To demonstrate a use case of Dim-R2, DC-RNNs were trained to reproduce neural spiketrains from wild-type mouse motor circuits during a reach-to-grab task (Fig. 20). The $y$ and $\hat{y}$ have shape (Random seed (RS), Cross validation folds (CV), Trial, Time, Neuron)=(3,3,78,249,252). Dim-R2 with $\mathcal{A}$=(RS, CV, Trial) reveals accuracy across Time and Neuron dimensions (Fig. 4a), where high values correspond to periods of strong trial variability (Fig. 3a,b,e,f). The DC-RNN captures trial variability well in DCN throughout most of the trial and in M1 after lift onset (100ms). In $y$, trial variability peaks at each neuron's maximal activation (Fig. 4b). Since neurons are sorted by peak activation time within each brain region, this appears as a diagonal band of high scores per region. This trial variability is also reflected in Dim-R2 with $\mathcal{A}$=(RS, CV, Time) (Fig. 4e, f), where the neurons with higher trial variability show vertical bands with higher scores for both $\hat{y}$ and $\bar{y}$.

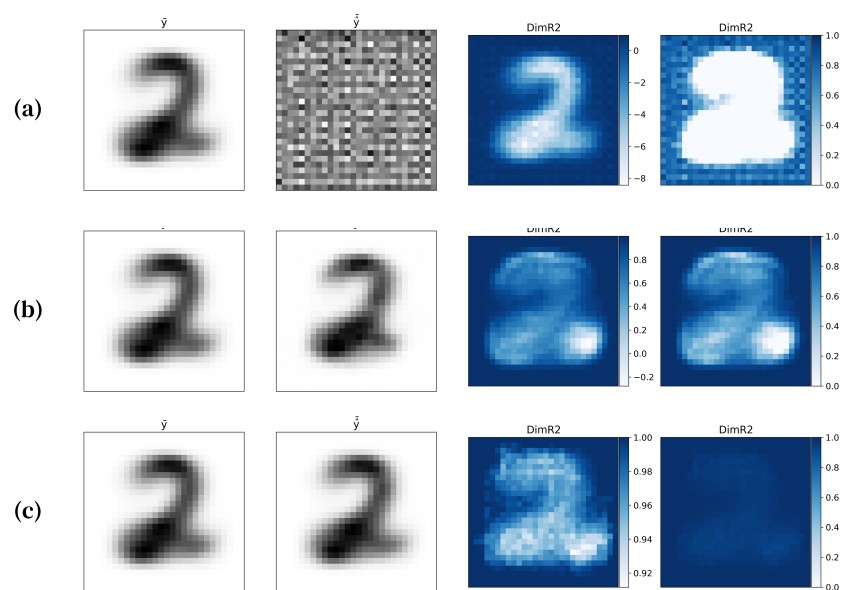

Figure 5: Dim-R2 shows dimensional view of regression accuracy for MNIST image reconstruction. Dim-R2 was measured with $\mathcal{A}$=(Data, Channel), $\mathcal{A}_{bias}$=$\mathcal{A}_{ref}$=(Width, Height). Columns show (from left to right): $\bar{y}$, $\hat{\bar{y}}$, Dim-R2 with a data min-max color scale, Dim-R2 with a fixed [0,1] color scale. Rows correspond to VAE training iterations: (a) 0 (before training), (b) 400, and (c) 33,640 (after early stopping)

Setting $\mathcal{A}_{ref}$=Time evaluates each neuron's prediction accuracy across time within individual trials. Dim-R2 reveals that some neurons in the DCN and late-onset neurons in M1 are well predicted by the DC-RNN (Fig. 4c, g), despite widespread trial variability in most neurons (Fig. 4b, f). Comparing Fig 4g and h shows that DC-RNN recreates activity in specific DCN and M1 neurons. Some noisy trials can also be identified in Fig. 4g, appearing as horizontal bands with low scores. Fig. 4h correctly yields a Dim-R2 of 0 when $\mathcal{A}$=Time as $\bar{y}$ are time averaged signals. Fig. 4d indirectly shows the underlying structure of the spiketrains, similar to Fig. 3d,g. These dimensional views reveal both the structure of the data $y$ and the prediction patterns of the DC-RNN, guiding the modeling process.

Dim-R2 can be used in any regression tasks to measure how well a model captures variation across different dimensions. To demonstrate additional use cases, VAEs were trained to reconstruct MNIST and CelebA images. For MNIST handwritten digit images, $y$ and $\hat{y}$ had shape (Data, Channel, Width, height)=(10000, 1, 28, 28). Dim-R2 was measured with $\mathcal{A}$=(Data, Channel) to leave (Width, Height) as the output shape, and $\mathcal{A}_{bias}$=$\mathcal{A}_{ref}$=(Width, Height) to capture spatial variance, which is intrinsic to images. This yields a normalized interpretable 2D Dim-R2 map (Fig. 5). Measuring Dim-R2 across VAE training iterations reveals which spatial features the model learns first. Initial prediction shows the structure of the data since $\mathcal{A} \neq \mathcal{A}_{bias}$ and is not 0 throughout (Fig. 5a). After 400 stochastic gradient descent updates (Fig. 5b), the VAE largely reconstructs the structure of the digit "2", although the lower-right tail remains poorly captures, suggesting the model first learns common spatial components shared across all digits. Fully trained VAE reconstructs the whole image at a high score (Fig. 5c).

For CelebA images, $y$ and $\hat{y}$ had shape (Data, Channel, Width, height)=(19962, 3, 128, 128). As these images had red, green, blue (RGB) channels, two types of Dim-R2 arguments were used to measure Dim-R2 across VAE training iterations. First, the same $\mathcal{A}$=(Data, Channel), $\mathcal{A}_{bias}$ = $\mathcal{A}_{ref}$=(Width, Height) were used to leave (Width, Height) as the output shape and capture spatial variance. Second, $\mathcal{A}$=Data, $\mathcal{A}_{bias}$ = $\mathcal{A}_{ref}$=(Width, Height) were used to leave (Channel, Width, Height) as the output shape that details per-channel Dim-R2, while still capturing the same spatial variance (Fig. 6). Initial prediction shows the structure of the data as $\mathcal{A} \neq \mathcal{A}_{bias}$ and is not 0 throughout (Fig. 6a). After 300 stochastic gradient descent updates (Fig. 6b), the VAE largely reconstructs the background area, and the hairs of the face, while it cannot reconstruct the pupil. Different channels show different accuracy patterns such as red channel Dim-R2 being lower around

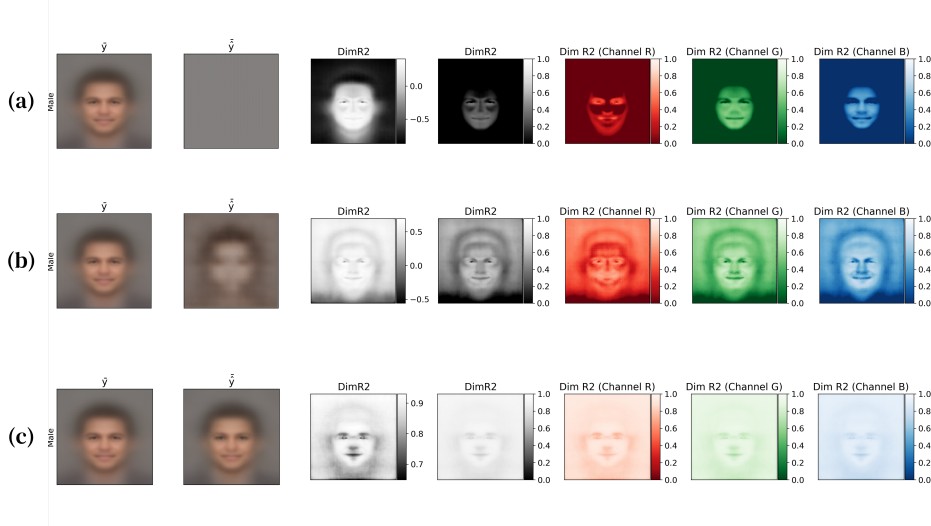

Figure 6: Dim-R2 shows dimensional view of regression accuracy for CelebA image reconstruction. Dim-R2 was measured under two parameter settings: Columns 3-4 use $\mathcal{A}$=(Data, Channel) and $\mathcal{A}_{bias}$=$\mathcal{A}_{ref}$=(Width, Height); Columns 5-7 use $\mathcal{A}$=Data, with the same $\mathcal{A}_{bias}, \mathcal{A}_{ref}$. Columns show (from left to right): $\bar{y}$, $\bar{\hat{y}}$, Dim-R2 with a data min-max color scale, Dim-R2 with a fixed [0,1] color scale, and per-channel Dim-R2 (Red, Green, Blue) a fixed [0,1] scale. Rows correspond to VAE training iterations: (a) 0 (before training), (b) 300, and (c) 39,280 (after early stopping)

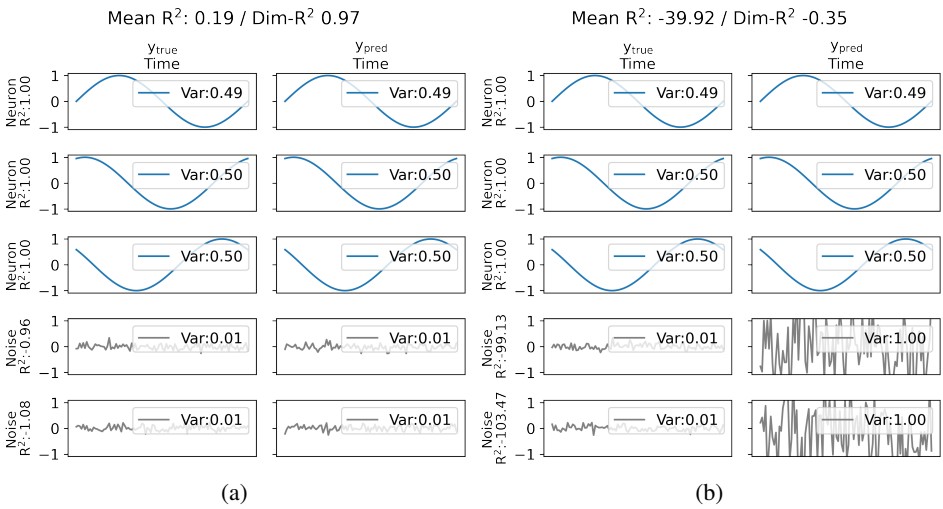

Figure 7: Example waveforms of $y$ and $\hat{y}$ with their corresponding mean R2 and Dim-R2 scores. (a) Noise channel variance: $y$=0.01, $\hat{y}$=0.01, (b) $y$=0.01, $\hat{y}$=1.00. Full variance combinations of $y$ and $\hat{y}$ noise channels are presented in Fig. 13.

the skin area compared to other channels. After full training, the VAE reconstructs the overall image at a high accuracy across the image (Fig. 6c). Full visualizations for all image classes across 16 training snapshots and the final model, for both MNIST and CelebA, are provided in the Supplemental File (A.7.2).

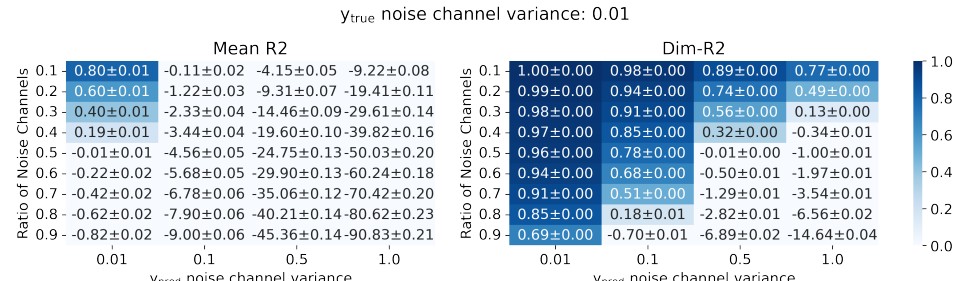

Figure 8: Dim-R2 highlights the presence of channels with high predictive accuracy in the presence of noisy channels. Scores were measured on simulated sinusoidal data (Fig. 7) across hyperparameter sweeps. Each entry shows the mean±standard deviation measured across 100 random repetitions.

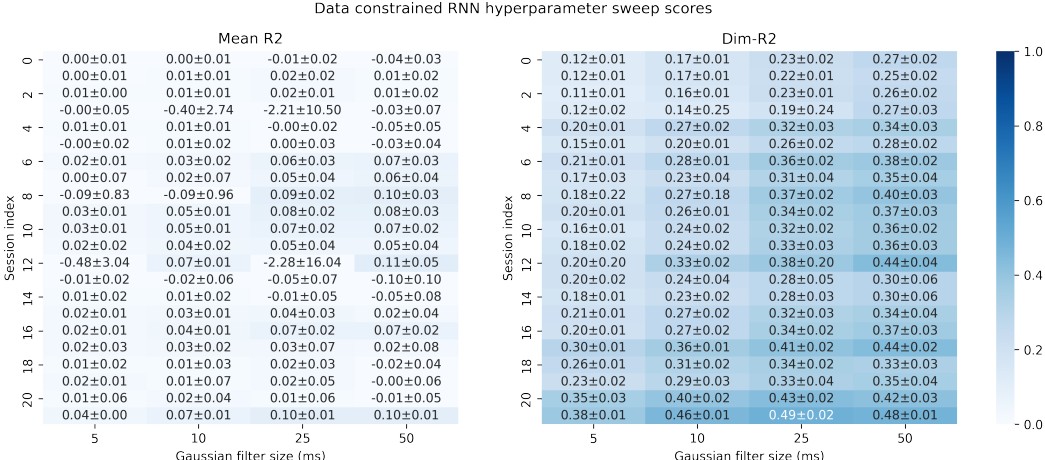

Figure 9: Dim-R2 highlights the presence of channels with high predictive accuracy in the presence of noisy channels. Scores were measured on data-constrained recurrent neural network predictions on neural spiketrains across hyperparameter sweeps. Each entry shows the mean±standard deviation.

## 4.2 DIM-R2 BETTER REFLECTS HIGH-ACCURACY CHANNELS THAN MEAN R2 IN THE PRESENCE OF NOISE CHANNELS

When $y$ has low variance, the normalizing term of R2 (TSS) becomes small which amplifies modest prediction errors into large negative values. This drags mean R2 to a negative value when multichannel $y$ contains noise channels. In contrast, Dim-R2 with $\mathcal{A} = \mathcal{D}$ (all dimensions) treats the collapsed dimensions as independent observations, allowing high variance informative channels to dominate the score. This reduces the effect of low variance noise channels, and better highlights the presence of high accuracy informative channels compared to mean R2.

To demonstrate how Dim-R2 better reflects the presence of high-accuracy channels, multichannel sinusoidal data (2D) was generated for $y$ and $\hat{y}$ (Fig. 7, 13). When the $y$ noise channel variance is low (0.01) compared to the signal channels (0.5), Dim-R2 yields significantly higher scores than mean R2 that highlight the presence of high accuracy channels (Fig. 8). This advantage decreases as the $y$ noise channel variance increases and eventually exceeds the signal channel variance (Fig. 14). Both mean R2 and Dim-R2 decrease as $\hat{y}$ noise channel variance increases. Additional suboptimal regression scores were also evaluated using the metrics in Table 1 for reference (Fig. 16-19).

To demonstrate the noise channel resilience use case of Dim-R2, Dim-R2 and mean R2 were measured on DC-RNN predictions of neural spiketrains across different recording sessions and Gaussian filter sizes used in preprocessing (Fig. 9). The scores gradually increase with larger filter sizes, as both $y$ and $\hat{y}$ become smoother. While this trend is visible in Dim-R2, it is less apparent in Mean R2

due to the low-variance noise channels that pull the score down. For instance, Sessions 3 and 12 at a 25ms filter size yield a moderately large negative mean R2 than other filter sizes, which is an outlier effect not observed in Dim-R2. This noise resilience can also be seen in each $y$ and $\hat{y}$ pairs from both trial-averaged and single trial neural activity (Fig. 20, Table 3). Thus, Dim-R2 allows users to identify the effects of hyperparameters on RNN performance without the confounding influence of noise.

## 5 DISCUSSION

When using Dim-R2 to assess accuracy across dimensions, careful consideration of $\mathcal{A}$, $\mathcal{A}_{bias}$, and $\mathcal{A}_{ref}$ is essential, especially when they differ. In such cases, a Dim-R2 score of 0 does not imply equivalent reference to a mean prediction but may instead reflect the data structure (Fig. 3d,h; 4d). To interpret these dimensional scores, it is important to compare against domain-specific controls for reference, such as time-averaged $\bar{y}$.

It is important to understand what Dim-R2 captures when reduced to a single value, as Dim-R2 and mean R2 weigh noise channels differently. With small-variance noise channels, mean R2 drops because all channels are weighed equally, while Dim-R2 concatenates all channels along one axis (Fig. 11) and is dominated by high-variance signals, under-weighing noise (Fig. 14a,b). This aligns with the interpretation of Dim-R2 as variance-weighted mean R2 (Appendix A.2, Fig. 15). Conversely, Dim-R2 decreases more in large variance noise channels because noise dominates the collapsed signal, while mean R2 under-weighs noise with equal weights. Thus, mean R2 is sensitive to small-variance noise channels while Dim-R2 is sensitive to high-variance noise channels. A less sensitive metric is better at detecting the presence of high-accuracy channels, whereas a more sensitive metric is better at measuring uniform accuracy across all channels, in which the metric sensitivity depends on noise variance. In many natural settings such as images, audio, and neural activity, signals have larger variance than noise, making Dim-R2 suitable for detecting high-accuracy channels.

The three benefits of Dim-R2 comes from applying a multidimensional generalization to a metric that average across per-channel measurements of the form $1 - Error/Normalization$. Metrics with this structure (Appendix A.1)–such as explained variance (a non-regression metric) and D2 absolute error (a suboptimal regression metric)–inherit the same three limitations of conventional R2. Extending these metrics using the same multidimensional generalization is straightforward and could yield the same three benefits demonstrated by Dim-R2.

## 6 CONCLUSION

We introduced Dim-R2, a regression metric that accepts data of arbitrary dimensionality, enables dimensional evaluation of regression accuracy, and highlights the presence of high-accuracy channels in the presence of noisy channels. Dim-R2 offers a bird's-eye view of regression accuracy and a noise-resilient score for reliable hyperparameter exploration. Its ability to reveal dimensional patterns offers guidance for model evaluation in multidimensional regression tasks such as image or signal reconstruction, commonly used in modeling (Huang et al., 2024; Yoo et al., 2024; Badrulhisham et al., 2024; Zador et al., 2023; Yang & Wang, 2020; Lin et al., 2023; Heckel et al., 2024; Ahmed et al., 2025; Zhang et al., 2023).

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

# A  APPENDIX

## A.1  COMMON REGRESSION METRICS

The following is the definitions of several commonly used regression metrics referenced in the Related Works section. While explained variance and correlation are not regression metrics, they are included as they give insights related to regression.

**Mean Squared Error (MSE).**

$$\text{MSE} = \frac{1}{N} \sum_{i=1}^{N} (y_i - \hat{y}_i)^2. \tag{7}$$

MSE penalizes large errors quadratically but is not normalized to the scale of the data and does not provide a reference baseline such as predicting the mean.

**Mean Absolute Error (MAE).**

$$\text{MAE} = \frac{1}{N} \sum_{i=1}^{N} |y_i - \hat{y}_i|. \tag{8}$$

MAE is robust to outliers compared to MSE but similarly lacks normalization and does not capture variance.

**D2 Absolute Error (D2-MAE).**

$$\text{D2-MAE} = 1 - \frac{\sum_{i=1}^{N} |y_i - \hat{y}_i|}{\sum_{i=1}^{N} |y_i - \bar{y}|}. \tag{9}$$

D2-MAE normalizes absolute error relative to the deviation of $y$ from its mean, producing a score in $(-\infty, 1]$ with interpretable anchors (1 = perfect prediction; 0 = equivalent to predicting $\bar{y}$). However, because it uses absolute error, it does not capture variance.

**Explained Variance (EV).**

$$\text{EV} = 1 - \frac{\text{Var}(y - \hat{y})}{\text{Var}(y)}. \tag{10}$$

Explained variance measures how well the fluctuations of $\hat{y}$ match those of $y$, but does not penalize additive bias (e.g., $\hat{y} = y + b$ yields EV $= 1$).

**Correlation (Corr).**

$$\text{Corr} = \frac{\text{Cov}(y, \hat{y})}{\sigma_y \sigma_{\hat{y}}}. \tag{11}$$

Pearson correlation quantifies linear co-variation between $y$ and $\hat{y}$ and is scale-invariant, but does not measure prediction error nor bias.

## A.2  PROPERTIES OF DIM-R2

Dim-R2 inherits the same bounds and invariances from conventional R2.

### A.2.1  BOUNDS OF DIM-R2

By definition,

$$Dim\text{-}R2 = 1 - \frac{RSS}{TSS} \tag{12}$$

where

$$RSS = \sum_{k \in \mathcal{A}} (y_k - \hat{y}_k)^2 \geq 0, \qquad RSS \in \mathbb{R}^{\mathcal{D} \setminus \mathcal{A}} \tag{13}$$

For each entry of the resulting tensor in $\mathbb{R}^{\mathcal{D} \setminus \mathcal{A}}$:

- $RSS = 0$ if and only if $\hat{y} = y$ along the axes in $\mathcal{A}$ yielding Dim-R2=1.

- $RSS = TSS$ when $\hat{y}$ along the axes in $\mathcal{A}$ equals $\bar{y}$ along the axes in $\mathcal{A}_{bias}$, yielding Dim-R2=0.

- $RSS > TSS$ yields negative values of Dim-R2.

Thus, for every entry,

$$Dim\text{-}R2 \in (-\infty, 1] \tag{14}$$

### A.2.2 INVARIANCES

Dim-R2 is invariant to affine transformations of both $y$ and $\hat{y}$. Let $a, b \in \mathbb{R}$, $a \neq 0$ and define

$$y' = ay + b, \qquad \hat{y}' = a\hat{y} + b. \tag{15}$$

RSS and TSS transform as

$$RSS(y', \hat{y}') = a^2\, RSS(y, \hat{y}), \qquad TSS(y') = a^2\, TSS(y). \tag{16}$$

Since the scaling factor $a^2$ cancels in the ratio, we obtain

$$Dim\text{-}R^2(y', \hat{y}') = 1 - \frac{RSS(y', \hat{y}')}{TSS(y')} = 1 - \frac{a^2 RSS(y, \hat{y})}{a^2 TSS(y)} = Dim\text{-}R^2(y, \hat{y}). \tag{17}$$

For the degenerate case of $a = 0$, the transformed target becomes constant,

$$y' = \hat{y}' = b, \tag{18}$$

so both RSS and TSS vanish:

$$RSS(y', \hat{y}') = TSS(y') = 0. \tag{19}$$

Following the standard convention for R2 when $TSS = 0$, the score is defined as 1 whenever the prediction equals the target. Therefore,

$$Dim\text{-}R^2(y', \hat{y}') = 1. \tag{20}$$

Thus, Dim-$R^2$ is invariant under any non-zero affine transformation.

### A.2.3 REDUCTION TO CONVENTIONAL R2

Dim-R2 can reduce to conventional R2 under specific argument choices.

1. **1D case**

   Let $y, \hat{y} \in \mathbb{R}^N$ with observation indices $i = 1, ..., N$. Let the full set of axes be $\mathcal{D} = \{1\}$.

   $$Mean\ R2 = 1 - \frac{\sum_{i \in} (y_i - \hat{y}_i)^2}{\sum_i (y_i - \overline{y})^2} \tag{21}$$

   For Dim-R2, choose $\mathcal{A} = \mathcal{A}_{ref} = \mathcal{A}_{bias} = \{1\}$.

   $$Dim\text{-}R2 = 1 - \frac{\sum_i (y_i - \hat{y}_i)^2}{\sum_i (y_i - \overline{y})^2} \tag{22}$$

2. **2D case reducing Dim-R2 to a single score**

   Let $y, \hat{y} \in \mathbb{R}^{N \times C}$, with observation indices $i = 1, ..., N$, and channel indices $c = 1, ..., C$. Let the full set of axes be $\mathcal{D} = \{1, 2\}$, where axis 1 is the observation dimension and axis 2 is the channel dimension.

   $$Var_c = \sum_i (y_{ic} - \overline{y}_c)^2 \tag{23}$$

$$Weighted\ Mean\ R2 = \frac{1}{\sum_c Var_c} \sum_c Var_c * R2_c \tag{24}$$

$$= \frac{1}{\sum_c Var_c} \sum_c Var_c [1 - \frac{\sum_i (y_{i,c} - \hat{y}_{i,c})^2}{Var_c}] \tag{25}$$

$$= \frac{1}{\sum_c Var_c} [\sum_c Var_c - \sum_c \sum_i (y_{i,c} - \hat{y}_{i,c})^2] \tag{26}$$

$$= 1 - \frac{\sum_c \sum_i (y_{i,c} - \hat{y}_{i,c})^2}{\sum_c Var_c} \tag{27}$$

For Dim-R2, choose $\mathcal{A} = \{1, 2\}$ and $\mathcal{A}_{ref} = \mathcal{A}_{bias} = \{1\}$ so that the reference variance (TSS) is computed along the observation axis.

$$\bar{y}_c = \frac{1}{N} \sum_i y_{i,c} \qquad \bar{y} \in \mathbb{R}^C \tag{28}$$

$$Dim\text{-}R2 = 1 - \frac{RSS}{TSS} \tag{29}$$

$$= 1 - \frac{\sum_{k \in \mathcal{A}} (y_k - \hat{y}_k)^2}{\frac{1}{|\mathcal{A}_{ref} \backslash \mathcal{A}|} \sum_{j \in \mathcal{A}_{ref} \backslash \mathcal{A}} \sum_{k \in \mathcal{A}} (y_{k,j} - \bar{y})^2} \tag{30}$$

$$= 1 - \frac{\sum_{k \in \mathcal{A}} (y_k - \hat{y}_k)^2}{\sum_{k \in \mathcal{A}} (y_k - \bar{y})^2} \tag{31}$$

$$= 1 - \frac{\sum_c \sum_i (y_{i,c} - \hat{y}_{i,c})^2}{\sum_c \sum_i (y_{i,c} - \bar{y}_c)^2} \tag{32}$$

$$= 1 - \frac{\sum_c \sum_i (y_{i,c} - \hat{y}_{i,c})^2}{\sum_c Var_c} \tag{33}$$

Thus, collapsing all dimensions ($\mathcal{A} = \mathcal{D}$) while measuring per-channel variance across the observation dimension ($\mathcal{A}_{bias} = 1$) reduces Dim-R2 to the conventional variance weighted mean R2. When there are no mean shifts in $y$ or $\hat{y}$ across the channel dimension (i.e. $\bar{y}_c = \mu$ for all $c \in \{1, ..., C\}$), choosing $\mathcal{A} = \mathcal{A}_{bias} = \mathcal{D}$ also yields same values for variance weighted R2 and Dim-R2.

3. **3D and higher case reducing Dim-R2 to a single score**

    The Conventional R2 score does not naturally extend to $y, \hat{y} \in \mathbb{R}^d$ for $d \geq 3$ in a way that preserves its interpretation, and therefore cannot be directly compared to Dim-R2.

## A.3 DIM-R2 DIAGRAMS

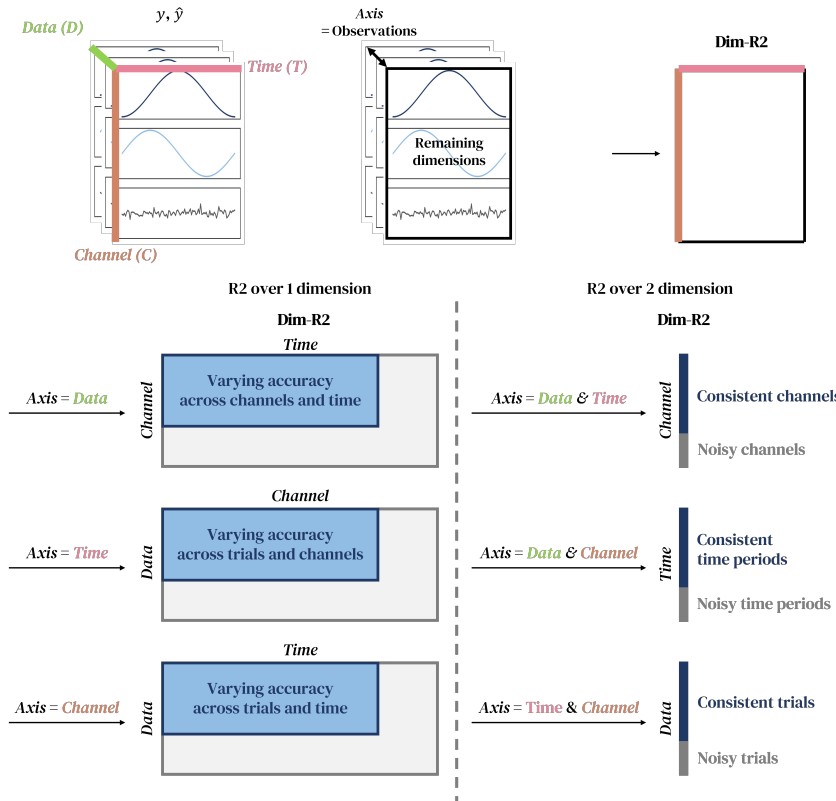

Figure 10: Dim-R2 presents rich patterns of prediction accuracy across designated dimensions (Axis).

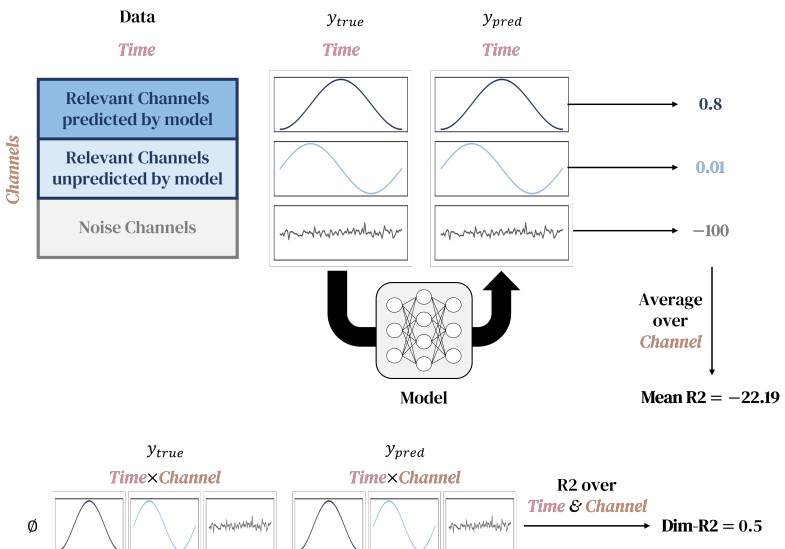

Figure 11: Dim-R2 is more resilient to noise channels than conventional mean R2 because high-variance signal channels dominate the influence of low-variance noise channels. As a result, Dim-R2 presents the existence of high accuracy channels among noise channels.

## A.4 DATA-CONSTRAINED RECURRENT NEURAL NETWORK PREDICTIONS ON MOUSE MOTOR CORTEX NEUROPIXEL RECORDINGS

### A.4.1 DATA

A total of 22 sessions were collected from four mice performing a reach-to-grab task (Sauerbrei et al., 2020; Guo et al., 2021; Levy et al., 2020). Neural activity (Spiketrains) was recorded simultaneously from the deep cerebellar nuclei (DCN), primary motor cortex (M1), Striatum, Thalamus using Neuropixels probes (Steinmetz et al., 2021). Spiketrains were recorded at 500Hz (2ms bin) where each value indicates the presence (1) or absence (0) of a spike. For each brain region, this results in a binary array of shape (time, neurons).

Hand kinematics were recorded using a camera at sampling rate of 500Hz, synchronized with the Neuropixels. The 3D hand coordinates (Units: mm) were extracted using Animal Part Tracker (Lee et al., 2020). Movement onset was defined as the time point when the hand position exited a predefined square region representing the resting state.

### A.4.2 PREPROCESSING

Preprocessing of spiketrains involved 5 steps: slicing it to the region of interest (100ms before to 400ms after movement onset), Gaussian filtering ($\sigma$=50ms), reordering neuron indices based on peak activity for interpretability (does not affect training), normalizing activity range to match the range of DC-RNN activation function, and stacking the neurons across brain regions. The train set metadata was used to process the validation and test set for neuron reordering and activity normalization stages.

### A.4.3 EXPERIMENT SETUP

The DC-RNNs were trained with 5 by 3 fold nested k-fold cross validation with 3 different random seeds, resulting in $5 \times 3 \times 3 = 45$ experiments per condition. Each train, validation, test split were stratified-split with respect to the number of reach success trials, to balance the data characteristics. The random seeds affected model weight initialization and cross validation splits.

The DC-RNN was a vanilla RNN where the number of hidden neurons matched the number of recorded neurons in the spiketrain data. The activation function was tanh. The Adam optimizer

(Kingma & Ba, 2014) was used to train the DC-RNN (Werbos, 1990) with learning rate of 1e-3 and batch size of 32. Early stopping stopped training by measuring Dim-R2 every 100 updates with patience value of 100.

DC-RNN weights were initialized using He initialization, a uniform distribution $\mathcal{U}(-\sqrt{\frac{1}{hidden\_size}}, \sqrt{\frac{1}{hidden\_size}}$, which is the default in PyTorch (Paszke et al., 2017). The DC-RNN was implemented with PyTorch (Paszke et al., 2017). The learning rate and batch size were selected after initial parameter sweep with learning rate of 1e-2, 1e-3, and batch size of 16, 32.

The $y$ and $\hat{y}$ were aggregated (Yoo et al., 2025) from predictions of different DC-RNN sweeps under the same experiment conditions, to evaluate the experiment condition. Resulting $y$ and $\hat{y}$ shapes were (Random seed, Cross validation folds, Trial, Time, Neuron).

### A.5 IMAGE RECONSTRUCTION USING VARIATIONAL AUTOENCODERS

#### A.5.1 DATA

**MNIST.** The MNIST handwritten digits dataset consists of grayscale images with shape (Channels, Width, Height) = (1, 28, 28). The pixel values were normalized to [0,1]. The training and test sets contain 60,000 and 10,000 images, respectively. Only the test set was used for evaluating Dim-R2. Although MNIST includes 10 digit classes, these labels were not used for training and were used only for class-wise visualization of Dim-R2.

**CelebA.** The CelebA face dataset contains 3-channel RGB images originally of size (178, 218), which were resized to (128, 128) for our experiments, yielding images of shape (Channels, Width, Height) = (3, 128, 128). The pixel values were normalized to [0,1]. The training and test sets contain 162,770 and 19,962 images, respectively, and the test set was used for evaluating Dim-R2. Each image also includes 40 binary attributes in one-hot format. For visualization, we selected the eight attributes that had the most balanced True/False counts: `Smiling`, `Attractive`, `Mouth_Slightly_Open`, `High_Cheekbones`, `Wearing_Lipstick`, `Heavy_Makeup`, `Male`, and `Wavy_Hair`. These attributes were not used during VAE training but were used for attribute-wise visualization of Dim-R2.

#### A.5.2 EXPERIMENTAL SETUP

The VAEs (Pu et al., 2016; Doersch, 2016; Pinheiro Cinelli et al., 2021) were trained in a single instance with the provided train and test dataset. The VAE was a Convolutional neural network with the following architecture described in Table 2, with number of filters $n_f = 10$ and a latent dimension size $n_z = 100$. The loss function was the following equation:

$$\mathcal{L}_{VAE} = \lambda_{KL}\mathcal{L}_{KL} + \mathcal{L}_{recon} \tag{34}$$

$$\mathcal{L}_{KL} = \frac{1}{2}(\sigma_z + \mu_z - 1 - \log \sigma_z) \tag{35}$$

where $\mathcal{L}_{recon}$ was L1 loss for MNIST and Binary Cross Entropy with logits loss for CelebA. The $\lambda_{KL}$ was set to 5e-4. The reparametrization trick was only used during training, and only $\mu_z$ was used during evaluation.

The Adam optimizer (Kingma & Ba, 2014) was used with learning rate of 1e-3 and batch size of 64. Early stopping stopped training by measuring mean absolute error every 40 updates with patience value of 50. The final $y$ and $\hat{y}$ shapes were (Data, Channels, Width, Height).

| Module | Architecture Details |
|---|---|
| **ConvBlock** | Conv2d(in_channels, out_channels, kernel_size, stride, padding) 
 ReLU 
 BatchNorm2d(out_channels) |
| **CNN Encoder** | Input: (Channels, H, W) 
 ConvBlock(Channels, $n_f$, k5, s2, p2) 
 ConvBlock($n_f$, $2n_f$, k5, s2, p2) 
 ConvBlock($2n_f$, $4n_f$, k3, s2, p1) 
 ConvBlock($4n_f$, $6n_f$, k3, s2, p1) 
 Flatten 
 Linear($\cdot$, 512) 
 ReLU 
 Linear(512, $2 \times n_z$) 
 Outputs: $\mu_z, \log \sigma_z$ |
| **CNN Decoder** (MNIST) | Linear($n_z, 7 \times 7 \times 6n_f$) 
 ReLU 
 Unflatten to $(6n_f, 7, 7)$ 
 ConvTranspose($6n_f \to 4n_f$, k3, s2, p1, op1), ReLU 
 ConvTranspose($4n_f \to$ out_ch, k5, s2, p2, op1) 
 Output: (Channels, 28, 28) |
| **CNN Decoder** (CelebA) | Linear($n_z, 4 \times 4 \times 6n_f$) 
 ReLU 
 Unflatten to $(6n_f, 4, 4)$ 
 ConvTranspose($6n_f \to 4n_f$, k3, s2, p1, op1), ReLU 
 ConvTranspose($4n_f \to 2n_f$, k3, s2, p1, op1), ReLU 
 ConvTranspose($2n_f \to 2n_f$, k5, s2, p2, op1), ReLU 
 ConvTranspose($2n_f \to n_f$, k5, s2, p2, op1), ReLU 
 ConvTranspose($n_f \to$ Channels, k5, s2, p2, op1) 
 Output: (Channels, 128, 128) |

Table 2: VAE encoder and decoder architectures used for MNIST and CelebA. $n_f$ = Number of filters, $n_z$ = latent dimension size, k = kernel size, s = stride, p = padding, op = output padding.

### A.6 SYNTHETIC SINUSOIDAL DATASET EXPERIMENT FIGURES

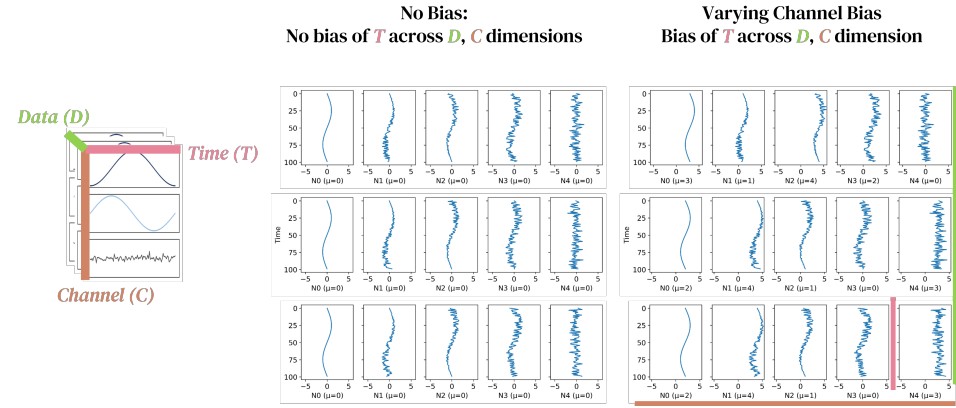

Figure 12: Two sinusoidal datasets ($y, \hat{y} \in \mathbb{R}^{D \times T \times C}$) with time-varying noise and added time biases. Different time biases are added which introduces variance across different dimensions, which can be captured by Dim-R2. Color-coded lines in the bottom right indicate each dimension type.

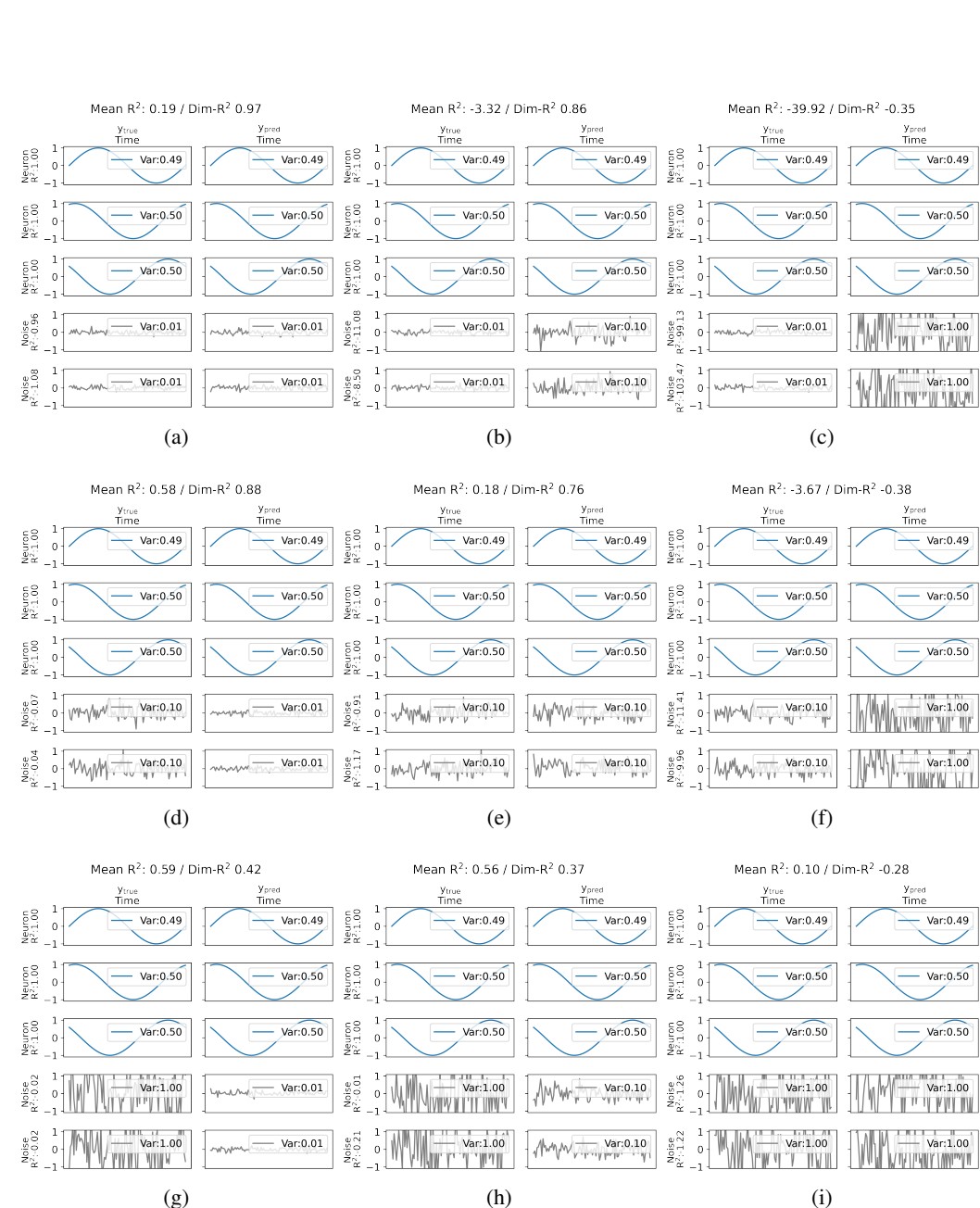

Figure 13: Example schematic waveforms of $y$ and $\hat{y}$ with their corresponding mean R2 and Dim-R2 scores. Each example shows sine waves with varying phases phases across 5 channels, where 2 channels have been replaced with Gaussian noise of specified variance. Rows share the same $y$ noise channel variance; columns share the same $\hat{y}$ noise channel variance. Legends show variances.

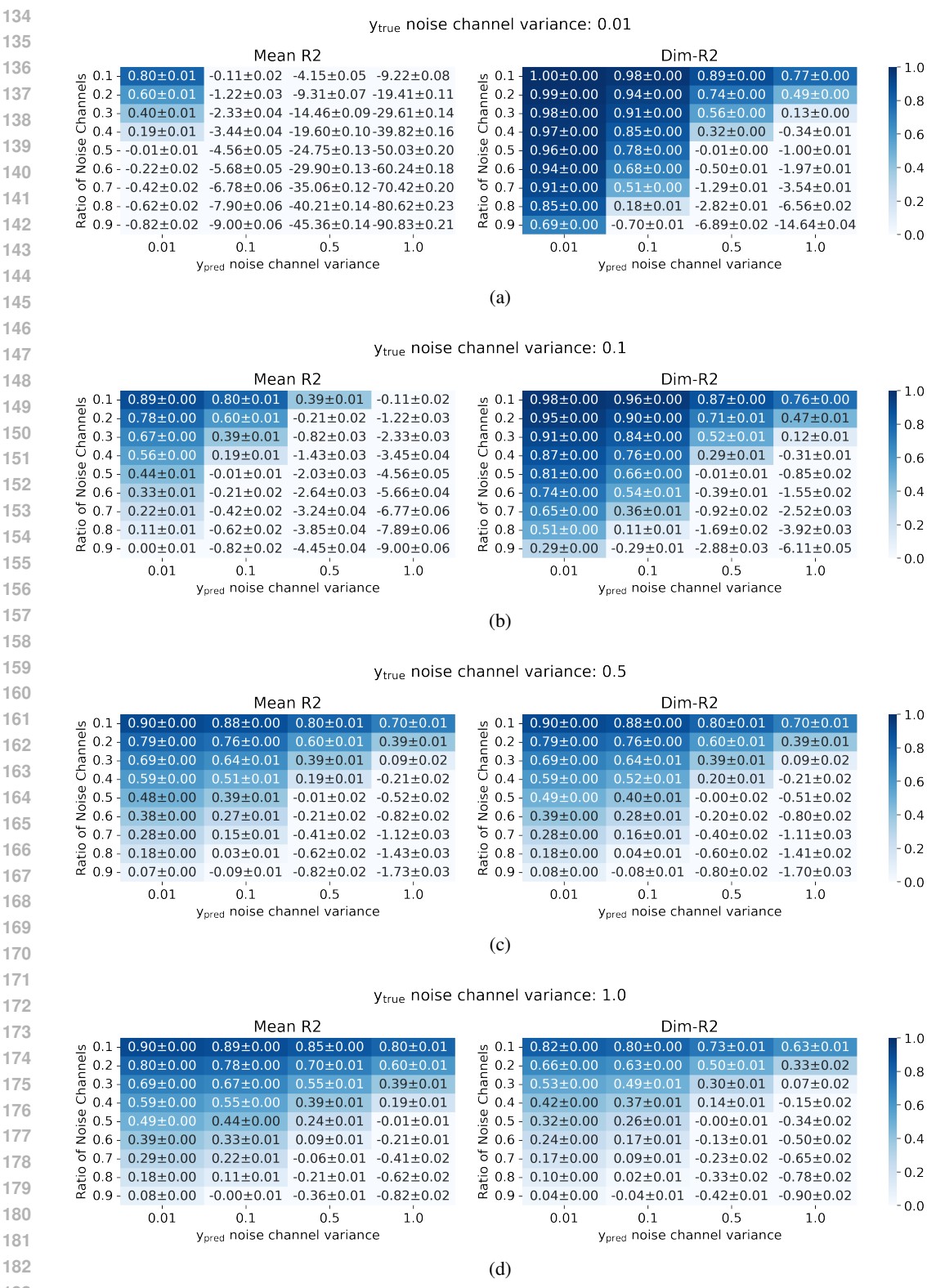

Figure 14: Dim-R2 highlights the presence of channels with high predictive accuracy in the presence of noisy channels. Scores were measured on simulated sinusoidal data (Fig. 7) across hyperparameter sweeps. Note in (c) that mean R2 and Dim-R2 show similar scores when $y$ noise channel variance equals the signal variance of 0.5 (Fig. 13). Each entry shows the mean±standard deviation across 100 random repetitions.

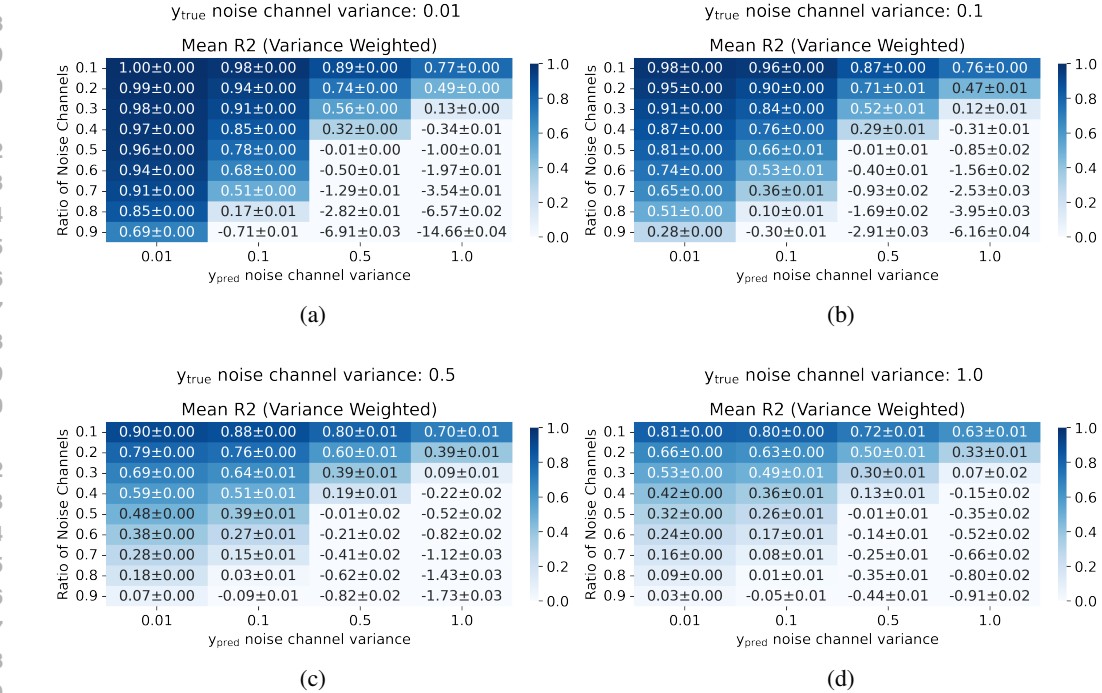

Figure 15: Variance weighted mean R2 scores measured on simulated sinusoidal data (Fig. 7) across hyperparameter sweeps. They match Dim-R2 scores (Fig. 14) because there were no mean shifts in $y, \hat{y}$ across the channel dimension. Each entry shows the mean±standard deviation across 100 random repetitions.

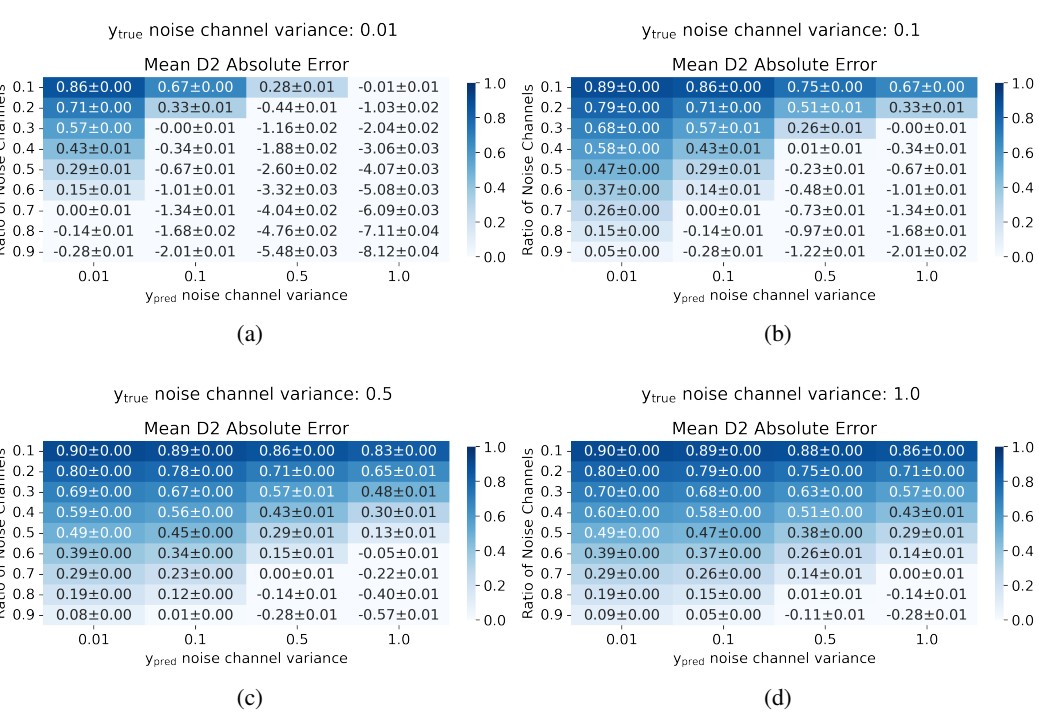

Figure 16: Mean D2 absolute error scores measured on simulated sinusoidal data (Fig. 7) across hyperparameter sweeps. They exhibit the same noise-sensitivity issues as mean-R2, whereas Dim-R2 remains stable under these conditions (Fig. 14). Each entry shows the mean±standard deviation across 100 random repetitions.

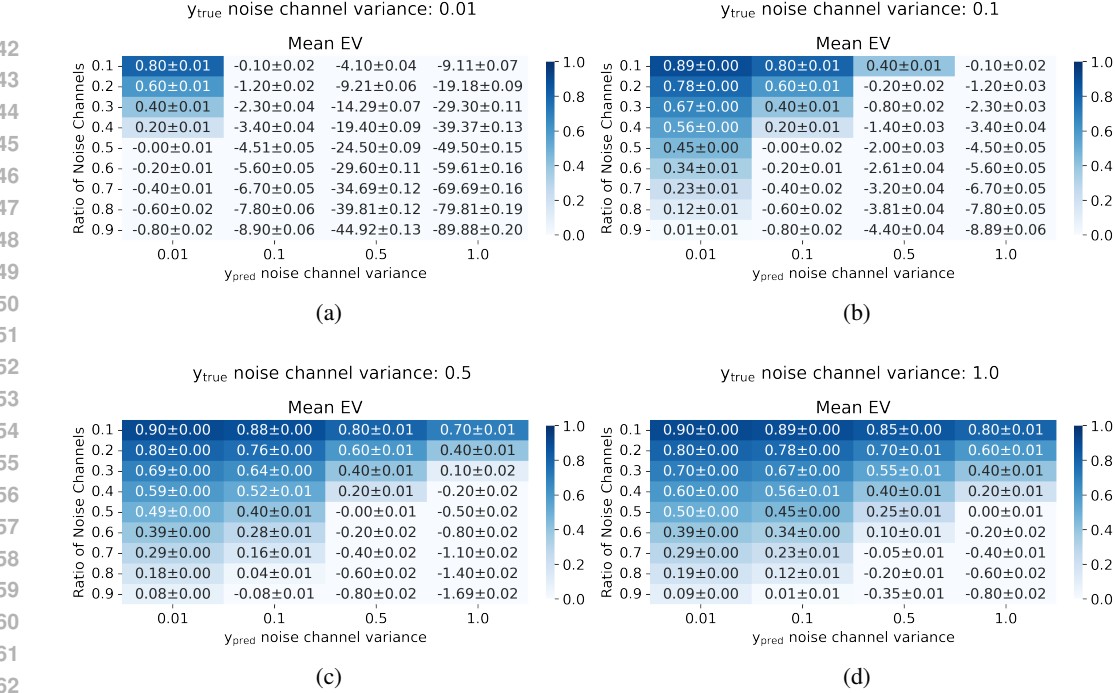

Figure 17: Mean explained variance scores measured on simulated sinusoidal data (Fig. 7) across hyperparameter sweeps. They exhibit the same noise-sensitivity issues as mean-R2, whereas Dim-R2 remains stable under these conditions (Fig. 14). Note the scores are identical to mean R2 in Fig. 14 because $E(y) = E(\hat{y})$ for $y, \hat{y}$ (Fig. 7). Each entry shows the mean±standard deviation across 100 random repetitions.

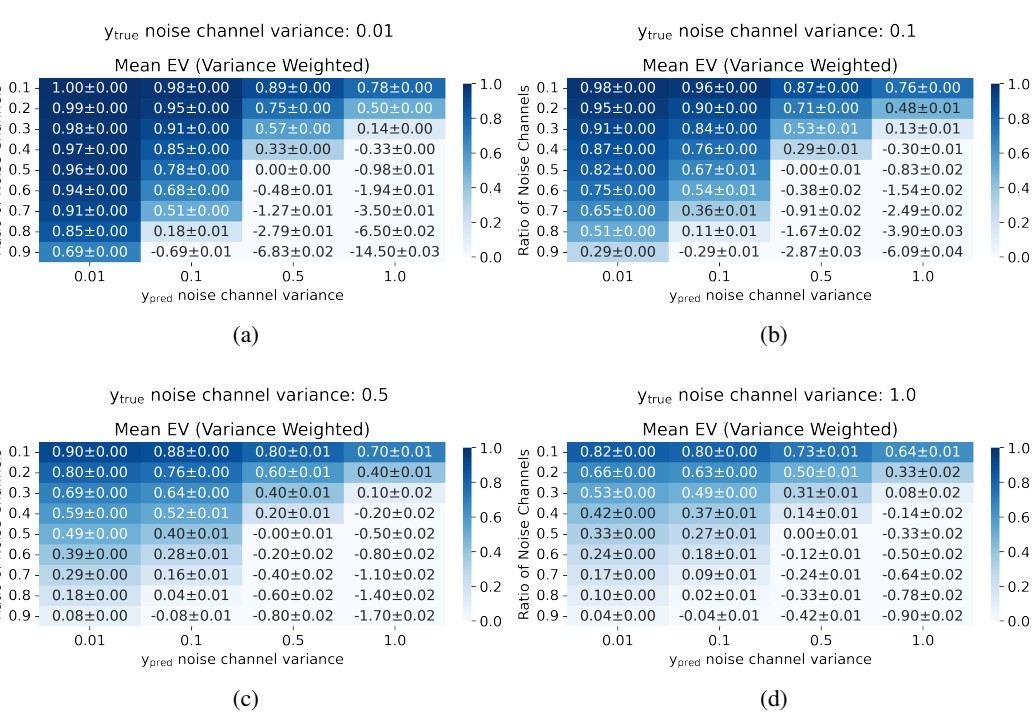

Figure 18: Mean explained variance scores measured on simulated sinusoidal data (Fig. 7) across hyperparameter sweeps. Note the scores are identical to Variance weighted mean R2 in Fig. 15 because $E(y) = E(\hat{y})$ for $y, \hat{y}$ (Fig. 7). Each entry shows the mean±standard deviation across 100 random repetitions.

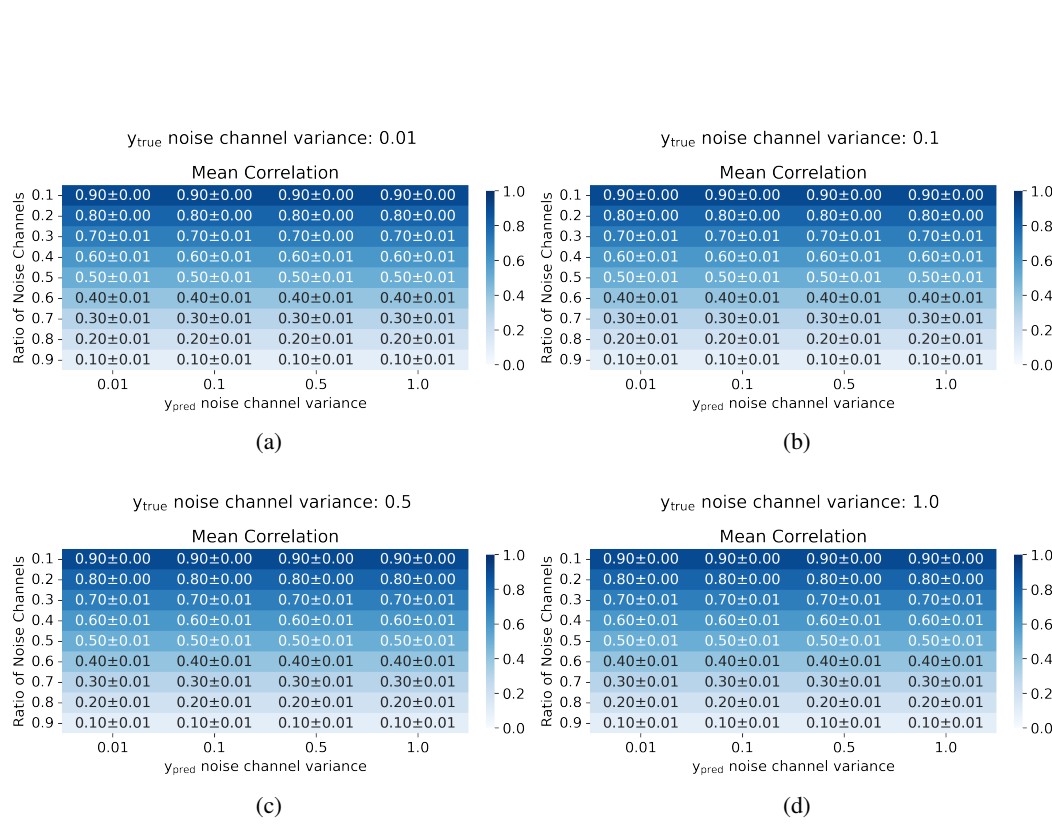

Figure 19: Mean correlation scores measured on simulated sinusoidal data (Fig. 7) across hyper-parameter sweeps. Correlation does not account for varying levels of $y$ and $\hat{y}$ noise variance, and therefore is not a regression metric. Each entry shows the mean±standard deviation across 100 random repetitions.

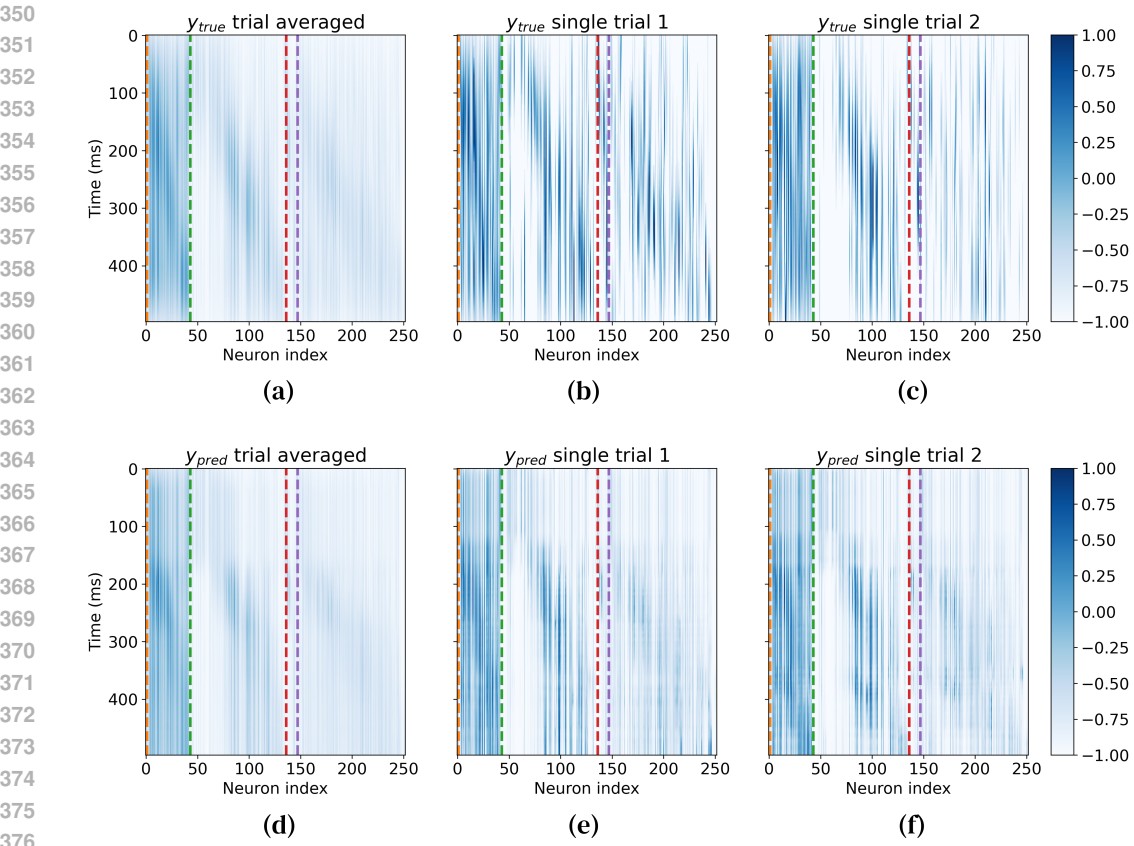

(a)                          (b)                          (c)

(d)                          (e)                          (f)

Figure 20: Examples of $y$ and $\hat{y}$ from DC-RNN trained to reproduce neural activity. This single session contains 78 trials, with 42, 93, 11, and 106 neurons from DCN, M1, Striatum, and Thalamus, respectively. This session corresponds to Session index 21 with a 50ms Gaussian filter in Fig. 9. Dashed lines separate neurons by brain region (from left): DCN (orange), M1 (green), Striatum (red), Thalamus (purple). (a) & (d) Trial averaged activity, (b) & (e) Single trial example 1, (c) & (f) Single trial example 2. (a)-(c) $y$, (d)-(f) $\hat{y}$.

|  | Trial averaged (a) & (d) | Single trial 1 (b) & (e) | Single trial 2 (c) & (f) |
|---|---|---|---|
| Dim-R2 | 0.92 | 0.44 | 0.57 |
| Mean R2 | 0.57 | -0.28 | -0.41 |

Table 3: Dim-R2 and Mean R2 measured on $y$ and $\hat{y}$ of Fig. 20. Dim-R2 yields higher scores than mean R2 when the $y$ and $\hat{y}$ are similar, even in the presence of noise.

## A.7 SUPPLEMENTAL FILES

### A.7.1 ALL DIM-R2 ARGUMENT COMBINATIONS ON SINUSOID DATA

### A.7.2 DIM-R2 MEASURED ACROSS ALL TRAINING ITERATIONS AND IMAGE CLASSES FOR MNIST AND CELEBA

### A.7.3 PYTHON IMPLEMENTATION OF DIM-R2

## A.8 THE USE OF LARGE LANGUAGE MODELS (LLMS)

The Large Language Model ChatGPT (https://chatgpt.com/) was used only to aid writing under strict author supervision.

