# OpenReview forum: "A dimensional R2 regression metric"
_ICLR.cc/2026/Conference — Submitted to ICLR 2026_

### Official Review · Reviewer_oZZX · 2025-10-18

**Soundness:** 2
**Presentation:** 1
**Contribution:** 1
**Rating:** 2
**Confidence:** 3

**Summary:**

The paper proposes Dimensional R² (Dim‑R²), a generalization of the coefficient of determination that: (1) works with arbitrarily high‑dimensional regression targets by collapsing user‑selected axes, (2) produces multidimensional maps of predictive accuracy instead of a single scalar, and (3) is less sensitive to low‑variance noise channels than mean per‑channel R². The method is specified via three arguments — Axis, Axis_ref, and Axis_bias — and computes RSS and TSS with broadcasting to preserve the remaining dimensions (Equations 3–6; schematic in Figure 3, p. 3). The paper demonstrates properties on (a) synthetic sinusoidal data with controlled noise and bias and (b) a data‑constrained RNN trained to reproduce mouse spiking activity during a reach‑to‑grab task. It highlights how Dim‑R² surfaces structure across time, trial, and neuron axes that is obscured by the mean of per‑channel R², and it reports that Dim‑R² is more robust when some channels are near‑constant noise.

**Strengths:**

- For practitioners, the dimensional view in Figures 6–7 (pp. 6–7) shows informative accuracy structure (e.g., time‑localized predictability and region‑specific effects) that scalar R² cannot reveal. The paper explains how changing Axis_ref changes what “variance explained” means operationally.
- Simple to implement and deploy. The formulation reduces to computing RSS/TSS over collapsed axes and broadcasting the shapes; a NumPy reference implementation following scikit‑learn syntax is mentioned. This lowers integration friction in typical pipelines.

**Weaknesses:**

- Novelty is narrower than claimed. The paper states conventional R² is “limited to at most two-dimensional inputs” and must be averaged per channel, but widely used libraries already support multi‑output R² with options (raw_values, uniform_average, variance_weighted) and variance‑weighted aggregation across outputs. At minimum, this weakens the motivation that basic R² “cannot” operate beyond 2D. The submission’s real contribution is a systematic axis‑aware mapping and reference‑variance design, not multi‑output support per se. Additionally, the nonparametric extension of R² can deal with functional data.
- Comparative baselines are too thin. The paper compares Dim‑R² mostly to mean per‑channel R². It does not compare against explained variance (which explicitly addresses constant‑variance pathologies) nor against off‑the‑shelf variance‑weighted R² and raw per‑output R² summaries. These are natural, strong baselines for the stated robustness claims.
- External validity is narrow. The main real‑world case study is neural spiking data with an RNN, a specialized domain. It’s unclear how Dim‑R² behaves on standard multi‑target ML benchmarks (multi‑variate time‑series forecasting, multi‑task regression, multichannel sensor prediction) or how it interacts with common evaluation practices there.

**Questions:**

- The authors are encouraged to strengthen their empirical results, by adding stronger baselines and a failure‑mode table. Include variance‑weighted R², raw per‑output R², explained variance (finite vs non‑finite behavior), and correlation maps, and summarize when each metric is recommended. One can also compare with nonparametric R².
- For functional data, like time series, R² does not provide a meaningful explanation, because of dependence among data points. Would the extension of R² also suffer the same issue?

---

> ### Author Response · Authors · 2025-11-29
>
> We appreciate the reviewer’s precise feedback to strengthen our manuscript. Here is the list of questions raised by the reviewer and our explanations with following modifications in the main text.
>
> 1. Dim-R2’s contribution regarding input/output dimensionality
>
> We appreciate the reviewer’s feedback to clarify our contribution regarding dimensionality. Conventional R2 is limited to 1 or 2D inputs, and can output 1 or 2D input based on averaging methods. With flexible axis-mapping as the reviewer mentioned, Dim-R2 removed the limit of the input as well as the output. We have clarified this in the “Introduction section” and “Section 3.1 REGRESSION METRICS: CONVENTIONAL AND DIMENSIONAL R2”.
>
> 2. Other regression metrics
>
> We appreciate the reviewer’s feedback to extend the metrics for comparison regarding noise-sensitivity. The additional baselines clarify the contribution of Dim-R2.
>
> Variance weighted R2 is a natural baseline to compare as it is another method of dealing with per-channel R2 averaging. In fact, Variance weighted R2 is a reduced form of Dim-R2 under certain argument selection, which we have described in the Appendix (A.2.3 REDUCTION TO CONVENTIONAL R2). Thus, measuring Variance weighted R2 resulted in same values with Dim-R2, under certain conditions which we have described in the Appendix (Fig. 15)
>
> We also evaluated several other metrics (D2 absolute error, Explained Variance, Correlation, Fig. 16-19) which may be informative but are not a suitable regression metric and thus not comparable to Dim-R2 nor conventional R2. Explained variance is similar to R2 but does penalize bias between $y$ and $\hat{y}$. Correlation measures co-fluctuation rather than predictive accuracy. D2 absolute error is normalized and has reference values to interpret but it cannot capture variance. These properties make them informative but conceptually distinct from a regression metric.
>
> To address the reviewer’s concern, we added a dedicated “Related Works” section describing the criteria that makes R2 as the standard metric for regression. We have also described how other metrics such as explained variance, correlation, D2 absolute error may be informative but are not a suitable regression metric, in the Related works and Appendix (A.1 COMMON REGRESSION METRICS).
>
> Finally, we note that metrics such as conventional R2, explained variance, D2 absolute error share a similar structure of 1-Error/Normalization and also average across channels in 2D input. Thus, they inherit the same three limitations of conventional R2. They could benefit from the same multidimensional generalization that we apply to R2, alleviating the same limitation of input dimension limit, sensitivity to noise, and creating a dimensional view. We have added this as a future work direction in the Conclusion section.
>
> 3. More applications
>
> We appreciate the feedback to include additional applications to strengthen the practical use cases of Dim-R2. The dimensional view provided by Dim-R2 applies to any multidimensional regression task. Thus, we incorporated a simple image-reconstruction example using MNIST and CelebA dataset. We measured Dim-R2 throughout training to illustrate which spatial features a VAE learns across stochastic gradient descent iterations.
>
> 4. Failure modes
>
> We appreciate the reviewer’s question regarding Dim-R2 failure modes (overstating or understating performance). Dim-R2 and mean R2 have different weighting schemes, and users should be aware of these differences. We have described this discretion in the Conclusion section.
>
> In summary, Dim-R2 and mean R2 differ in how they weigh noise channels:
>
> - Small-variance noise channels: Mean R2 is hypersensitive; Dim-R2 is stable
> - Large-variance noise channels: Dim-R2 is hypersensitive, mean R2 is stable.
>
> In many natural settings such as images, audio, and neural activity, true signal channels typically exhibit much larger variance than noise. With the goal of identifying the existence of high accuracy channels, Dim-R2 may be preferred over mean-R2.
>
> 5. Using regression metrics for functional data
>
> We appreciate the reviewer’s feedback on the problems of using regression metrics for autocorrelated functional data. We would like to highlight that this study is focused on regression metrics in machine learning context, not regression models or statistical measures. Regression metrics measure pointwise accuracy and do not measure model consistency diagnostics. We have added a short explanation that R2 regression score is a metric used for regression tasks in Machine Learning context, in the Related Works Section.
>
> ---
>
> Also, we would like to ask the reviewer to clarify one of their questions:
>
> 1. From “Additionally, the nonparametric extension of R² can deal with functional data.”, what does the reviewer mean by nonparametric extension of R2? Also, what does functional data mean in this context?

---

### Official Review · Reviewer_rB8j · 2025-10-30

**Soundness:** 2
**Presentation:** 2
**Contribution:** 2
**Rating:** 2
**Confidence:** 4

**Summary:**

This paper proposed the "Dimensional R2 (Dim-R2)" score, an extension of the conventional R2 (coefficient of Determination) Metric. The author claims that the conventional R2, as the gold standard, has three key limitations:
- is limited to at most two-dimensional data (is defined for 1D input and is averaged across channels for 2D input)
- reduces model performance to a single scalar, hiding the insight into how accuracy varies across data dimensions
- is highly sensitive to low-variance noise channels in multi-channel regression tasks, yielding large and uninterpretable negative values

The Dim-R2 metric aims to solve those limitations by "flattening" selected dimensions of multidimensional data into independent observations and computing the single and standard R2 score, while retaining the shape of the remaining dimensions. So that the author claim the Dim-R2 will provide three key benefits:
- accepts regression input of "arbitrary dimensionality"
- provides a "multidimensional view of prediction accuracy", helping modelers identify patterns in both data and models
- will be more "resilient to noise" than the conventional Mean R2 score, yielding a more robust score with high-variance (information) channels outweigh low-variance (noise) channels.

**Strengths:**

The paper has reasonable motivation: the common practice of Mean R2 scores for multidimension data could be sensitive to noise and can hide performance patterns. Providing a "dimensional view" of accuracy can be useful for specific applications such as neuroscience.

**Weaknesses:**

1. The synthetic sinusoidal dataset is a simple sine wave with added noise. A mouse neural recording dataset is quite specialized. Without additional datasets, from popular modalities like images, audio, and text, existing experiments are insufficient to demonstrate the generalizability and broad applicability of the proposed metric.

2. The paper frames the problem as "Mean R2 is limited, so here is Dim-R2." But It is highly unlikely that no other similar solutions for conventional (Mean) R2 have been proposed. This lack of a thorough literature review and citations on existing metrics makes the paper's contributions isolated. This can also be seen from the fact that, most of the cited references are not from the ML field; I am not sure if ICLR is the right venue for this paper.

3. It will be better if there is theoretically analysis why or how to introduce the Axis, Axis_ref, and Axis_bias, as well as the relationship between them, and theoretically analysis why or how the high-variance (informative) channels outweigh low-variance (noisy) ones.

**Questions:**

See weaknesses.

---

> ### Author Response · Authors · 2025-11-29
>
> We appreciate the reviewer’s precise feedback to strengthen our manuscript. Here is the list of questions raised by the reviewer and our explanations with following modifications in the main text.
>
> 1. More applications
>
> We appreciate the feedback to include additional applications to strengthen the practical use cases of Dim-R2. The dimensional view provided by Dim-R2 applies to any multidimensional regression task. Thus, we incorporated a simple image-reconstruction example using MNIST and CelebA dataset. We measured Dim-R2 throughout training to illustrate which spatial features a VAE learns across stochastic gradient descent iterations.
>
> 2. Citations and literature review
>
> We also appreciate the feedback on a thorough literature review on regression. To address the reviewer’s concern, we added a dedicated “Related Works” section describing the criteria that makes R2 as the standard metric for regression. We have also described how other metrics such as explained variance, correlation, D2 absolute error may be informative but are not a suitable regression metric, in the Related works and Appendix (A.1 COMMON REGRESSION METRICS).
>
> We appreciate the reviewer’s feedback on including more machine learning (ML) related literature to strengthen the relevance of Dim-R2 for a broader ML audience. We have included several more regression literature in our new “Related works section” that studied both standard (1-2 dimensional data) or multidimensional (3+ dimensional data) regression.
>
> 3. Heuristics on Dim-R2 arguments and its characteristics
>
> We appreciate the Reviewer’s feedback on clarifying on Axis selection heuristics. We revised the Dim-R2 Axis descriptions in the main text (Section 3.1. REGRESSION METRICS: CONVENTIONAL AND DIMENSIONAL R2) to clarify the roles of Axis, Axis_ref, and Axis_bias. This updated text now provides intuition for how each axis choice influences which variability is measured, helping users select axes that capture the specific structure they wish to evaluate and visualize.
>
> Also, we have included the Supplemental File (A.8.1 ALL DIM-R2 ARGUMENT COMBINATIONS ON SINUSOID DATA; All Dim-R2 argument combinations on sinusoid data.zip) containing all combinations of Axis, Axis_ref, and Axis_bias evaluated on the toy neural data with Varying Channel Bias.

---

### Official Review · Reviewer_yn9R · 2025-10-30

**Soundness:** 2
**Presentation:** 2
**Contribution:** 1
**Rating:** 4
**Confidence:** 3

**Summary:**

The paper proposes Dim-R$^2$, a new regression evaluation metric that extends the conventional R$^2$ score to handle data of any dimensionality. The standard R$^2$ is limited due to three reasons: restriction to 1–2D data, reduction of model performance to a single scalar, and sensitivity to low-variance noise channels. Dim-R$^2$ addresses these issues by flattening selected data dimensions and computing specified variance reference scores (via adjusting Axis_ref, Axis_bias), enabling a multidimensional view of model accuracy and reducing noise sensitivity. The authors validated Dim-R$^2$ using both synthetic sinusoidal data and an RNN trained on mouse neural recordings, showing that it effectively reveals structured prediction patterns across time, neurons, and trials. Moreover, results demonstrate that Dim-R$^2$ better highlights high-accuracy channels in noisy datasets than mean R$^2$, providing an interpretable metric for regression analysis.

**Strengths:**

- The paper identifies weaknesses in the conventional R² metric and motivates the need for an improved measure for regression accuracy.
- Dim-R² is straightforward and intuitive, and generalizes to multidimensional data without deviating much from the familiar interpretation of R².

**Weaknesses:**

**The presentation could be largely improved:**

- When reading some paragraphs in this paper, I often had to scroll up and down to look for figures paired with the text. For example (but not limited to these examples), equations (3-6) should be located closer to Figure 3. Figures 4 and 5 should appear on the same or adjacent page as Section 2.2.1.
- Although the authors clarified that “Throughout this paper, the terms dimensions and axes, data and trials, and channels and neurons are used interchangeably”, frequent changes of these terms in the paper have negatively influenced the coherence of the paper reading.
- There is no content in Section 2.2.2. I assume that this is some typo, and subsequent subsections should belong to the title of Section 2.2.2?

**Some experimental details are unclear to me and need more clarification:**

- For the synthetic sinusoidal dataset: If I understand correctly, $y_{true}$ and $y_{pred}$ with no added bias, differ only in the Gaussian noise added to them. What was the process of sampling the waveforms across trials? What are the parameters of the Gaussian noise distribution? How does the noise increase/decrease over time? Do the independent noises for $y_{true}$ and $y_{pred}$ follow the same Gaussian distribution? In terms of the Varying Bias case, is the bias also randomly assigned to $y_{true}$ across trials and neurons?
- For the real-world dataset: What are the actual shapes (Random seeds, Validation folds, Test batch, Time, Channels) of the dataset used for computing Dim-R$^2$?
- What is the Axis_bias used in the experiments in Figure 6? Since Axis_bias $\subset$ Axis_ref, I assume that Axis_bias=Trial in Figure 6(a,b,e,f) and Axis_bias=Time in Figure 6(c,d,g,h)?
- These experimental details should have appeared in the Appendix for reproduction purposes.

**Concerning the impact/significance of this paper for the broader machine learning community,** the authors demonstrate Dim-R$^2$ on only one real dataset in the area of neural science. Although the authors argue in the Conclusion section that Dim-R$^2$ can serve as a better metric of regression accuracy in general AI areas, the citations there are mostly related to neural science. Could the authors provide more examples of how Dim-R$^2$ might exhibit an advantage over the original R$^2$ for tasks in other areas? Examples may include, but are not limited to the following: robotic imitation learning, autoencoder reconstruction loss, and the DDPM objective of diffusion models. Moreover, I do not see a related work section in the paper. Could the authors also discuss other metrics evaluating regression accuracy, in addition to R$^2$?

**Minor Issues**:

- Conventionally, the coefficient of determination is denoted $R^2$ (R-squared). And in Figures 1, 2, 3, 8, the authors wrote $R^2$ but $R2$ everywhere else.

**Questions:**

Please refer to the weaknesses.

---

> ### Author Response · Authors · 2025-11-29
>
> We appreciate the reviewer’s precise feedback to strengthen our manuscript. Here is the list of questions raised by the reviewer and our explanations with following modifications in the main text.
>
> 1. Presentation
>
> We appreciate the feedback on the presentation. We have modified the figure placements and reorganized the Materials and Methods Section, including the previous misaligned section “DATA-CONSTRAINED RECURRENT NEURAL NETWORK PREDICTIONS ON MOUSE MOTOR CORTEX NEUROPIXEL RECORDINGS”. We have also changed the vocabularies of the dimensions to be (data, trial, channels) in the main text and toy examples, and corresponding dimension names for different applications.
>
> 2. Description of details
>
> We appreciate the reviewer’s feedback on describing experiment details. We have added more experimental details about noise parameters in the dataset description of “Section 3.2.1 SYNTHETIC SINUSOIDAL DATASET” and data shape in the “Section 4.1 DIM-R2 PROVIDES A DIMENSIONAL VIEW OF REGRESSION ACCURACY”.
>
> 3. More applications
>
> We appreciate the feedback to include additional applications to strengthen the practical use cases of Dim-R2. The dimensional view provided by Dim-R2 applies to any multidimensional regression task. Thus, we incorporated a simple image-reconstruction example using MNIST and CelebA dataset. We measured Dim-R2 throughout training to illustrate which spatial features a VAE learns across stochastic gradient descent iterations.
>
> 4. Citations
>
> We appreciate the reviewer’s feedback on including more machine learning (ML) related literature to strengthen the relevance of Dim-R2 for a broader ML audience. We have included several more regression literature in our new “Related works section” that studied both standard (1-2 dimensional data) or multidimensional (3+ dimensional data) regression.
>
> 5. Other regression metrics
>
> We appreciate the reviewer’s feedback to extend the metrics for comparison regarding noise-sensitivity. The additional baselines clarify the contribution of Dim-R2.
>
> Variance weighted R2 is a natural baseline to compare as it is another method of dealing with per-channel R2 averaging. In fact, Variance weighted R2 is a reduced form of Dim-R2 under certain argument selection, which we have described in the Appendix (A.2.3 REDUCTION TO CONVENTIONAL R2). Thus, measuring Variance weighted R2 resulted in same values with Dim-R2, under certain conditions which we have described in the Appendix (Fig. 15)
>
> We also evaluated several other metrics (D2 absolute error, Explained Variance, Correlation, Fig. 16-19) which may be informative but are not a suitable regression metric and thus not comparable to Dim-R2 nor conventional R2. Explained variance is similar to R2 but does penalize bias between $y$ and $\hat{y}$. Correlation measures co-fluctuation rather than predictive accuracy. D2 absolute error is normalized and has reference values to interpret but it cannot capture variance. These properties make them informative but conceptually distinct from a regression metric.
>
> To address the reviewer’s concern, we added a dedicated “Related Works” section describing the criteria that makes R2 as the standard metric for regression. We have also described how other metrics such as explained variance, correlation, D2 absolute error may be informative but are not a suitable regression metric, in the Related works and Appendix (A.1 COMMON REGRESSION METRICS).
>
> Finally, we note that metrics such as conventional R2, explained variance, D2 absolute error share a similar structure of 1-Error/Normalization and also average across channels in 2D input. Thus, they inherit the same three limitations of conventional R2. They could benefit from the same multidimensional generalization that we apply to R2, alleviating the same limitation of input dimension limit, sensitivity to noise, and creating a dimensional view. We have added this as a future work direction in the Conclusion section.

---

### Official Review · Reviewer_dcXe · 2025-11-10

**Soundness:** 2
**Presentation:** 1
**Contribution:** 2
**Rating:** 2
**Confidence:** 3

**Summary:**

This paper introduces the dimensional R2 regression metric (Dim-R2), which extends the conventional mean R2 score. In particular, it aims to address three key limitations of R2: (1) its limited scope for 2D data, (2) compression of accuracy notions into a single scalar, and (3) sensitivity to low-variance noise channels. Subsequently, the proposed Dim-R2 incorporates a multi-dimensional view of accuracy and is more robust than mean per-channel R² under low-variance noise channels. This proposal is supported by empirical evidence on synthetic sinusoidal studies and data-constrained RNNs predicting mouse Neuropixels activity.

**Strengths:**

* The paper presents a relevant objective to extend the gold standard (R2 score) to have a multi-dimensional view. It identifies three limitations of conventional R2 and aims to address them with the extension.
* The formulation is simple and straightforward, with formulas given in Equations 3-6 and accompanying visualizations.

**Weaknesses:**

* **Underdeveloped Theory:** The proposed Dim-R2 metric is an intuitive extension of the R2 score from classical statistical theory, but only the formulas are presented. More theoretical analysis (bounds/equality, invariances, reduction to classical R2, etc.) would build confidence that Dim-R2 behaves predictably beyond the shown cases. I found the further theory to be both doable and necessary because the work serves to extend a fundamental statistic; however, there’s no properties section establishing basic guarantees.

* **Presentation issues:** Some of the notations in Section 2 do not look mathematically professional, and the subscripts are difficult to read and confusing. I recommend dropping the “true” part and use y, \hat{y}, \bar{y} for ground-truth, predicted, and mean labels. Additionally, text phrases Axis, Axis_ref, and Axis_ref-Axis are used as sets, which is not professional. Please consider using rigorous set notations and clean the notations with care.

* **Limited Empirical Evaluation:** The baseline comparisons are very narrow, namely the authors only compare it with the traditional R2. This significantly impacts the claimed novelty and robustness of the metric. Stronger baselines (weighted R2, correlation, mutual-information-type measures etc.) that can capture learned data variation should be considered to support the claims. Additionally, from my understanding, the Axis sets should be manually selected; this somewhat limits the practical value of this metric. If we have insights on how to define the set, we probably already know about some data structure and are aware of the noise channels.

* **Revising Structure:** With the above recommended additions, I would suggest moving some visualizations that help explain Dim-R2 and experiment settings to the Appendix. Currently, these occupy too much space of the main text and do not significantly help with audiences' understanding given the text explanations.

While I appreciate the idea of extending R2, I recommend “reject” for the lack of both theoretical guarantees and empirical depth in this paper.

**Questions:**

Most questions are raised in the Weakness section. Additionally,

1. Can you provide more concrete ablation studies on the Axis selection? This could help with selection heuristics.

2. Are there any tangible counterexamples where Dim-R² could overstate performance (failure modes)? It is saying that this method necessarily has flaws, but it is good to know both sides of the story.

---

> ### Author Response · Authors · 2025-11-29
>
> We appreciate the reviewer’s precise feedback to strengthen our manuscript. Here is the list of questions raised by the reviewer and our explanations with following modifications in the main text.
>
> 1. Theory
>
> We appreciate the reviewer’s feedback to strengthen the theoretical aspects of Dim-R2. We have added the “Properties of Dim-R2” in the appendix that describes the “Bounds of Dim-R2”, “Invariances to affine transformation”, and “Reduction to conventional R2”.
>
> 2. Presentation & Structure
>
> We very much appreciate the feedback on a cleaner math notation. We have modified the math notations at Fig. 1, 3 along with the main text explaining the math variables. We have also moved the diagrams Fig. 1, 2, 5 to the appendix to better focus the main text.
>
> 3. Additional Empirical Evaluation
>
> We appreciate the reviewer’s feedback to extend the metrics for comparison regarding noise-sensitivity. The additional baselines clarify the contribution of Dim-R2.
>
> Variance weighted R2 is a natural baseline to compare as it is another method of dealing with per-channel R2 averaging. In fact, Variance weighted R2 is a reduced form of Dim-R2 under certain argument selection, which we have described in the Appendix (A.2.3 REDUCTION TO CONVENTIONAL R2). Thus, measuring Variance weighted R2 resulted in same values with Dim-R2, under certain conditions which we have described in the Appendix (Fig. 15)
>
> We also evaluated several other metrics (D2 absolute error, Explained Variance, Correlation, Fig. 16-19) which may be informative but are not a suitable regression metric and thus not comparable to Dim-R2 nor conventional R2. Explained variance is similar to R2 but does penalize bias between $y$ and $\hat{y}$. Correlation measures co-fluctuation rather than predictive accuracy. D2 absolute error is normalized and has reference values to interpret but it cannot capture variance. These properties make them informative but conceptually distinct from a regression metric.
>
> To address the reviewer’s concern, we added a dedicated “Related Works” section describing the criteria that makes R2 as the standard metric for regression. We have also described how other metrics such as explained variance, correlation, D2 absolute error may be informative but are not a suitable regression metric, in the Related works and Appendix (A.1 COMMON REGRESSION METRICS).
>
> Finally, we note that metrics such as conventional R2, explained variance, D2 absolute error share a similar structure of 1-Error/Normalization and also average across channels in 2D input. Thus, they inherit the same three limitations of conventional R2. They could benefit from the same multidimensional generalization that we apply to R2, alleviating the same limitation of input dimension limit, sensitivity to noise, and creating a dimensional view. We have added this as a future work direction in the Conclusion section.
>
> 4. Ablation studies to help Axis selection heuristics
>
> We appreciate the Reviewer’s feedback on clarifying on Axis selection heuristics. We revised the Dim-R2 Axis descriptions in the main text (Section 3.1. REGRESSION METRICS: CONVENTIONAL AND DIMENSIONAL R2) to clarify the roles of Axis, Axis_ref, and Axis_bias. This updated text now provides intuition for how each axis choice influences which variability is measured, helping users select axes that capture the specific structure they wish to evaluate and visualize.
>
> Also, we have included the Supplemental File (A.8.1 ALL DIM-R2 ARGUMENT COMBINATIONS ON SINUSOID DATA; All Dim-R2 argument combinations on sinusoid data.zip) containing all combinations of Axis, Axis_ref, and Axis_bias evaluated on the toy neural data with Varying Channel Bias.
>
> 5. Counterexamples where Dim-R2 can overstate performance
>
> We appreciate the reviewer’s question regarding Dim-R2 failure modes (overstating or understating performance). Dim-R2 and mean R2 have different weighting schemes, and users should be aware of these differences. We have described this discretion in the Conclusion section.
>
> In summary, Dim-R2 and mean R2 differ in how they weigh noise channels:
>
> - Small-variance noise channels: Mean R2 is hypersensitive; Dim-R2 is stable
> - Large-variance noise channels: Dim-R2 is hypersensitive, mean R2 is stable.
>
> In many natural settings such as images, audio, and neural activity, true signal channels typically exhibit much larger variance than noise. With the goal of identifying the existence of high accuracy channels, Dim-R2 may be preferred over mean-R2.

---

### Meta-Review · Area_Chair_pgeT · 2026-01-06

**Summary:**

This paper proposes Dimensional R² (Dim-R²), an extension of the conventional R² regression metric for arbitrarily high-dimensional outputs.

The paper received three reject and one borderline reject review.
Reviewers appreciated the motivation and visualizations.

Reviewers raised several concerns, including:
- underdeveloped theory
- limited evaluation
- limited guidance on usage and design
- missing expanded analysis, failure modes

**Reviewer Concerns:**

Concerns addressed include the expanded theory.

Concerns that remain outstanding include a broader scope of empirical evaluation (despite adding image reconstruction experiments, which seem tangential to the target setting of the R2 measure) and expanded analysis and failure modes (the explanation in the rebuttal reads very superficially).

Finally, several reviewers found issues of clarity and presentation significant.

**Reviewer Scores:**

dcXe - raise score (2) to 4
yn9R - keep score (4)
rB8j - keep score (2)
oZZX - keep score (2)

---

### Decision · Program_Chairs · 2026-01-26

Reject